# Rapid single-wavelength lightsheet localization microscopy for clarified tissue

Li-An Chu [1,2,10], Chieh-Han Lu [2,3,9,10], Shun-Min Yang[2], Yen-Ting Liu [3], Kuan-Lin Feng[4], Yun-Chi Tsai [3], Wei-Kun Chang[1], Wen-Cheng Wang[3], Shu-Wei Chang [3], Peilin Chen [3], Ting-Kuo Lee[2], Yeu-Kuang Hwu[2], Ann-Shyn Chiang [1,2,4,5,6,7,8]* & Bi-Chang Chen [1,3]*

Optical super-resolution microscopy allows nanoscale imaging of protein molecules in intact biological tissues. However, it is still challenging to perform large volume super-resolution imaging for entire animal organs. Here we develop a single-wavelength Bessel lightsheet method, optimized for refractive-index matching with clarified specimens to overcome the aberrations encountered in imaging thick tissues. Using spontaneous blinking fluorophores to label proteins of interest, we resolve the morphology of most, if not all, dopaminergic neurons in the whole adult brain ($3.64 \times 10^7 \mu m^3$) of *Drosophila melanogaster* at the nanometer scale with high imaging speed ($436 \mu m^3$ per second) for localization. Quantitative single-molecule localization reveals the subcellular distribution of a monoamine transporter protein in the axons of a single, identified serotonergic Dorsal Paired Medial (DPM) neuron. Large datasets are obtained from imaging one brain per day to provide a robust statistical analysis of these imaging data.

[1] Brain Research Center, National Tsing Hua University, 101, Section 2, Kuang-Fu Road, Hsinchu 30013, Taiwan. [2] Institute of Physics, Academia Sinica, 128 Sec. 2, Academia Road, Nankang, Taipei 11529, Taiwan. [3] Research Center for Applied Sciences, Academia Sinica, 128 Sec. 2, Academia Road, Nankang, Taipei 11529, Taiwan. [4] Institute of Systems Neuroscience, National Tsing Hua University, 101, Section 2, Kuang-Fu Road, Hsinchu 30013, Taiwan. [5] Department of Biomedical Science and Environmental Biology, Kaohsiung Medical University, 100, Shin-Chuan 1st Road, Sanmin Dist., Kaohsiung City 80708, Taiwan. [6] Institute of Molecular and Genomic Medicine, National Health Research Institutes, 35 Keyan Road, Zhunan, Miaoli County 35053, Taiwan. [7] Graduate Institute of Clinical Medical Science, China Medical University, 91, Hsueh-Shih Road, Taichug 40402, Taiwan. [8] Kavli Institute for Brain and Mind, University of California at San Diego, UC San Diego, 9500 Gilman Drive, La Jolla, CA 92093-0115, USA. [9] Present address: Department of Genetics and Complex Diseases, Harvard T H Chan School of Public Health, Boston, MA 02115, USA. [10] These authors contributed equally: Li-An Chu, Chieh-Han Lu. *email: aschiang@life.nthu.edu.tw; chenb10@gate.sinica.edu.tw

Exceeding the Abbe diffraction limit of optical microscopy has been one of the most exciting technological developments in life science. The unprecedented nanometric resolution achieved by several super-resolution techniques, including stimulated emission depletion microscopy[1], structured illumination microscopy[2], photo-activated localization microscopy[3] and stochastic optical reconstruction microscopy[4], has led to numerous breakthroughs in biology. Despite their successes, these techniques require the use of a high numerical aperture (NA) objective with short working distance which limited the observation depth to the vicinity of a coverslip surface[5,6], yielding relatively small imaging volumes compared to the size of most tissue specimens. Several research groups have attempted to modify the designs of localization microscopy techniques to accommodate larger imaging volumes[7–11]. Among these studies, a combination of ultra-thin Bessel[12] or lattice lightsheet[13] and point accumulation for imaging in nanoscale topography microscopy has achieved sub-100 nm resolution in samples with over 20-μm thickness[10]. Premature photo-bleaching was effectively suppressed by utilizing the inherent optical sectioning of lightsheet[9–11], whereby an oversampling localization density was attained[10,11]. Moreover, lightsheet illumination extends the imaging area away from the coverslip, thereby enabling the use of thicker specimens, such as neuromast organs and zebrafish embryos[10]. By choosing appropriate imaging parameters, the combination of lightsheet and localization microscopy can be used with living cell for tracking protein complex[11].

A more ambitious goal is to apply this technology and to perform three-dimensional (3D) imaging of subcellular features in large biological tissues. Due to photon scattering in these tissues, however, the quality of the image significantly degrades with observation depth, making super-resolution imaging much more difficult. To reduce the out-of-focus signal, genetic tools can be used to limit the expression of proteins within a small subset of neurons[14]. With mechanical sectioning, super-resolution imaging can, therefore, be carried out on 8-μm cryosections to reveal the subcellular features of neurons[14]. Because the structural integrity of a tissue is easily disrupted by mechanical sectioning[14], the challenge has remained in implementing such techniques to larger specimens or to map protein distributions within the whole brain. Two recent studies have used a mounting medium with a high refractive-index (RI) to capture super-resolution images of neurons and mRNA molecules from the whole fly brain using Airyscan[15] or Bessel Beam Structured Illumination Microscopy[16], respectively. The optical resolution in these studies nonetheless was still limited to greater than 100 nm. Further, the dehydration caused by organic solvents in the mounting medium likely altered the structural integrity of the samples and the chemical properties of the labeled fluorophores[16].

To address these issues, we use a water-based optical clearing agent to eradicate the RI mismatch arising from different cell morphologies and constituents within the fly brain, thereby minimizing light scattering and ensuring minimal perturbation of the sample structure. To break the depth of field (DOF) limit of detection objective with long working distance, the profile of the ultra-thin Bessel lightsheet convolves with the detection point spread function (PSF) to create a thin optical sectioning plane. Refined by post-processing based on the defocusing model, the signal-to-background ratio is improved[11], thus we are able to confined single-molecule localization within each optical section, and also to achieve super-resolved sub-100 nm lateral resolution throughout the entire sample. In addition, we use spontaneously blinking fluorophores HMSiR[17], which have been synthesized specially for super-resolution localization microscopy. By densely labeling the proteins of interest with spontaneously blinking fluorophores, we are able to map complex nanoscale structures within a considerable imaging volume using our lightsheet

localization microscopy. Our success in establishing a strategy for 3D super-resolution microscopy at nanometric resolution opens up exciting applications, including the quantification of synaptic protein densities through large brain volumes.

## Results

**Lightsheet localization microscopy for clarified tissue.** Our lightsheet localization microscopy for clarified tissue (LLM-CT) shares a similar optical path with lattice lightsheet microscopy (LLSM[13]; Fig. 1a–d and Methods). Instead of using a spatial light modulator to generate lightsheet in LLSM, the lightsheet was generated from a scanning Bessel beam created by a lens-axicon doublet, illuminated by a Gaussian laser beam[18] (Fig. 1b). Illuminated by varied Gaussian beam sizes, different ring-shaped focal patterns were formed, which were tuned to modulate the lightsheet length (Supplementary Fig. 1). The resulting ring pattern was further filtered by an annular ring mask placed on the Fourier plane conjugated to the back aperture of the customized excitation objective (Fig. 1a, c; N.A. = 0.5, working distance = 12.8 mm; Supplementary Fig. 2b)[19,20]. The principle of scanning Bessel beam lightsheet microscopy has been discussed in previous papers[19,20], and a detailed description of our system can be found in the methods section. The Bessel beam has been described as a self-reconstruction light beam that is particularly effective for penetrating a thick tissue[19,20]. In optically clarified tissues (Fig. 1d), the axicon-generated Bessel beam of over 200 μm (Fig. 1e, Supplementary Fig. 3i, o) is scanned through the imaging plane to form a time-averaged virtual lightsheet. The axial energy distribution of the illumination is mostly confined within a 0.55 μm thick layer which reduces premature bleaching of the fluorophore outside the imaging plane (Supplementary Fig. 3m, n). Volumetric imaging is conducted by scanning simultaneously the detection objective and the excitation lightsheet. Because the imaging volume created by scanning the lightsheet is relatively large in relation to the volume of a fly brain, whole-brain super-resolution can be reconstructed by stitching only four sub-volumes (Fig. 1f).

**Localization precision in optically cleared fly brains.** Traditional localization microscopy relies on photon-triggered quantum state switching of fluorophores either with exposure to a short wavelength or with high-power illumination leading to ground state depletion[3,4,21]. Use of an additional activation laser prolongs image acquisition time and potentially introduces more photo-bleaching. The high intensity illumination, on the other hand, will exert exceeding laser fluence to the surroundings near the imaging plane, which causes premature photo-bleaching and loss of localization density. To produce a localization microscope system with a moderate excitation power, we employed a spontaneously blinking fluorophore, HMSiR[17], which can be excited efficiently by a red-wavelength laser (λ = 637 nm) at a relatively low power density. This low excitation power requirement allows the imaging area in our set-up from a single exposure to be as large as $2 \times 10^4$ μm² (Fig. 2a).

To demonstrate our localization microscopy technique, a tissue sample maintained in phosphate-buffered saline (PBS) first was imaged with a water-based lightsheet system[11] (Supplementary Figs. 2a, 4). As depicted in Supplementary Fig. 4, the positions of fluorophores were identified unambiguously by using Thunder-STORM[22], and the reconstructed images demonstrated improved resolution. Due to tissue-induced aberrations, we observed an increase in lateral localization uncertainty in deeply located structures (Supplementary Fig. 4a). This increase in localization uncertainty resulted in a deterioration of the resolution of the reconstructed image and the increasing observed line width of dendritic structures, compared to images derived from shallower regions (Supplementary Fig. 4b). To improve this technique for

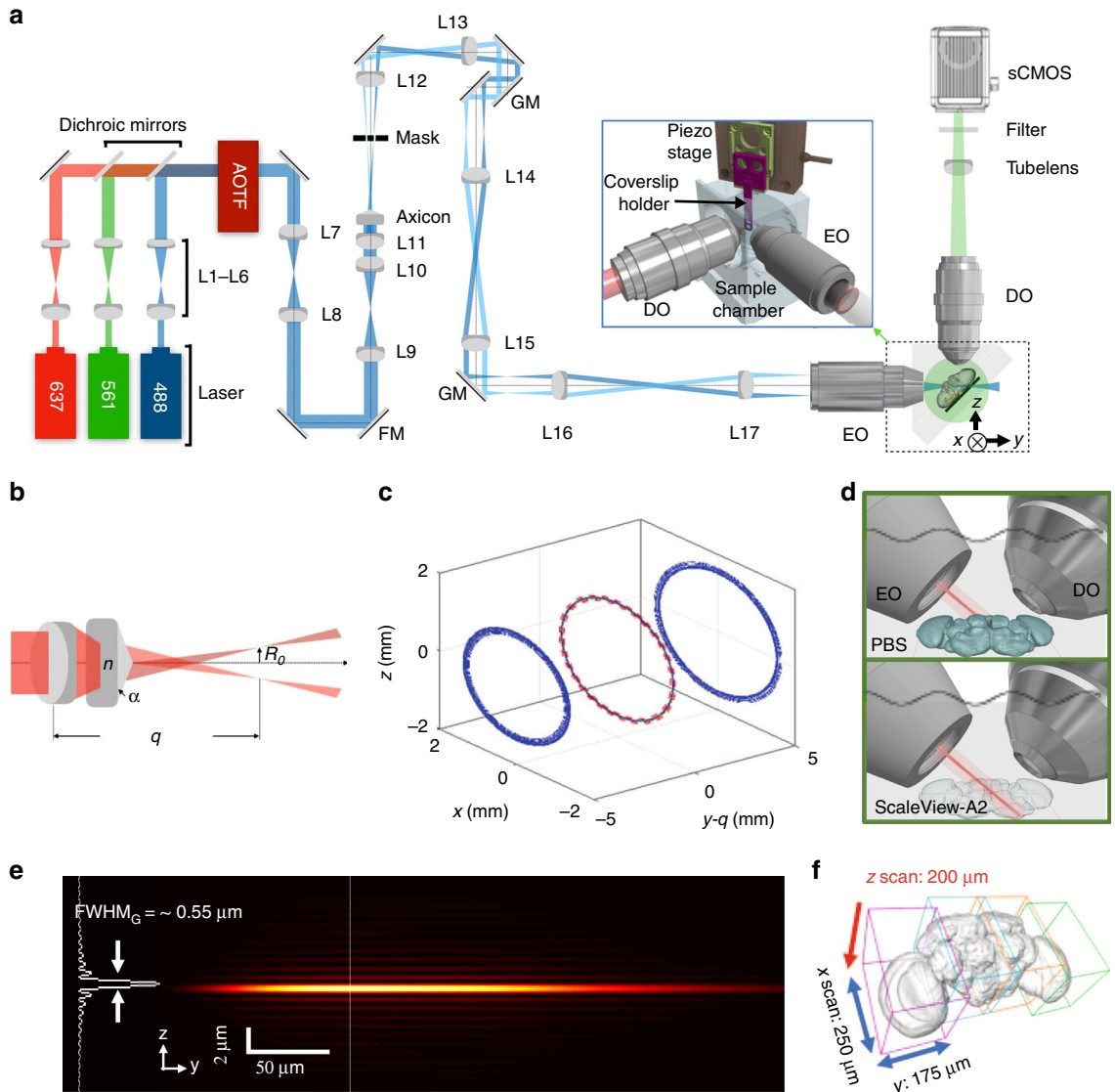

**Fig. 1** Lightsheet localization microscopy for clarified tissue. **a** The schematic configuration of the axicon based LLM-CT system. Inset: the geometry of relative positions between the two objective lenses and the specimen. AOTF, acoustic-optical tuneable filter; EO, excitation objective; DO, detection objective; GM, galvo mirror; FM, flip mirror, L, lens. Inset: detailed mechanical configuration of the imaging chamber. **b** Detailed illustrations of the Bessel beam generated using lens-axicon combination (L11 and axicon in **a**). With the convergent axicon with an apex angle $\alpha$ and refractive index $n$, the radius of the projected ring $R_O$ at the image plane of the lens (L11) can be written as $R_O = \alpha(n-1)q$. **c** Illuminated by a Gaussian laser beam profile (FWHM = 5 mm), the simulated intensity profiles of focused ring patterns at different positions along the optical axis of the focal lens, where the dashed red line indicates the ring pattern on the mask at the image plane ($y - q = 0$, $q$ is the image distance). **d** By using ScaleView-A2 to clarify the fly brain sample, the illumination can penetrate into deep tissue. **e** The simulated intensity profiles of the Bessel beams generated from the customized excitation objective in ScaleView-A2. The line profile indicates the intensity distribution along the lateral direction. **f** The whole-brain super-resolution image can be stitched by four sub-volumes scanned by LLM-CT. The dimension of each sub-volume is $250 \times 175 \times 200\ \mu m^3$.

imaging deep tissues, we integrated lightsheet with optical tissue clearing technology[23,24]. When the fly brain was cleared with neutral ScaleView-A2 solution (Methods)[25], the blinking signals observed in MZ19-Gal4-labeled olfactory neurons, located around 100 μm deep, became more distinguishable and had a better signal-to-noise ratio than those observed with PBS-based images (Fig. 2b–d, Supplementary Fig. 5). As we imaged through all HMSiR-labeled MZ19-Gal4 projection neurons, whose dendrites extend from the antennal lobe (AL) at the frontal surface of the brain to the axon terminals innervating the calyx of mushroom body (MB) at the posterior brain surface, we determined that our system was able to localize all blinking signals through an entire brain with minimum change in localization uncertainty and without mechanical sectioning (Fig. 2e, f).

**Whole-brain imaging using LLM-CT for HMSiR-labeled neurons.** LLM-CT allowed 3D nanoscopic imaging of MZ19-Gal4-labeled neurons. The superior quality of these images represents a substantial improvement over conventional lightsheet images acquired using the same system configuration (lightsheet microscopy for clarified tissue, LM-CT), including excitation wavelength and optics (Supplementary Fig. 6, Supplementary Movie 1). Due to the dense labeling of HMSiR (Fig. 3a), over 50 million molecules were localized in just 3 glomeruli on one side of antennal lobes with a lateral precision of ~20 nm over 60 μm depth in 1200 repeated imaging volumes (Fig. 3b; enlarged in Fig. 3c, Supplementary Movie 2). To determine a realistic acquisition time with a reliable statistical basis, we plotted the theoretical resolution with respect to time based on the structural features of our samples (Fig. 3d–g).

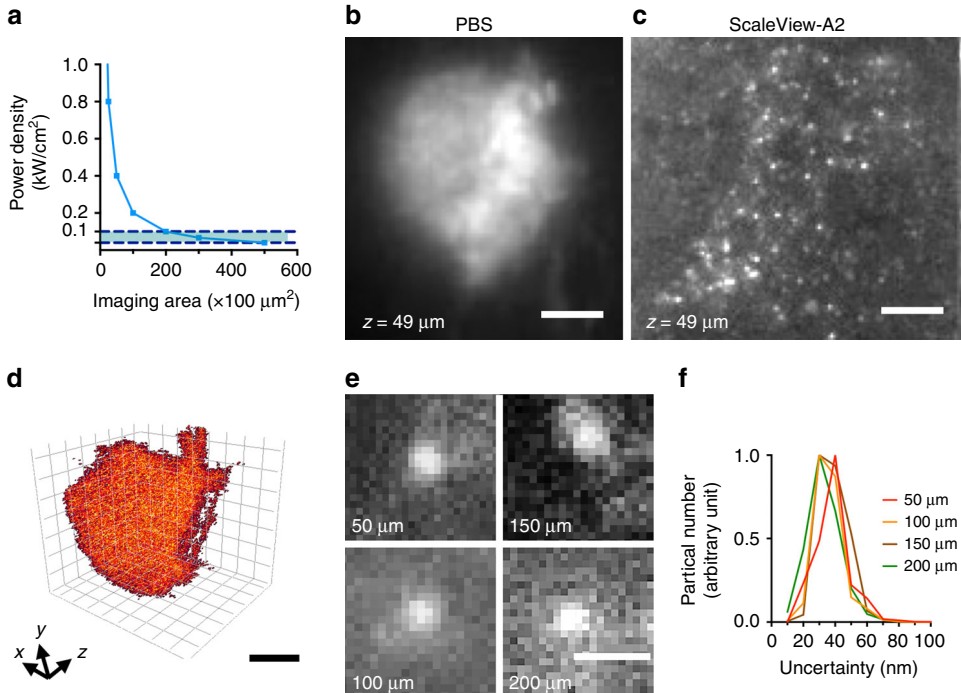

**Fig. 2** HMSiR blinking signal at different depths with tissue clearing. **a** The power density can be achieved by our system is plotted with respect to the imaging area that can be covered by lightsheet. The estimation is made by assuming all the power measuring at the back aperture of the excitation objective (20 mW) is transduced into the lightsheet and the imaging area is product of the axial length and the scanning range of the Bessel beam. The maximum axicon lightsheet coverage is 20,000 μm² (at the power density of 0.1 kW/cm²). Recommended power density ranges for HMSiR activation are indicated in blue. **b** Representative images of HMSiR blinking signal in phosphate-buffered saline (PBS) and **c** in Scalview-A2 at the same imaging depth (z = 49 μm) by imaging the HMSiR-labeled membrane GFP in olfactory projection neurons in *MZ19-Gal4 > UAS-mCD8::GFP* transgenic flies. Images were taken either under a ×25 water dipping objective lens in PBS or a ×25 objective lens specifically designed for use with ScaleView-A2. **d** Three-dimensional distribution of the signals can be localized in a single volumetric scan. **e** Representative HMSiR blinking events at four different imaging depths and **f** overall lateral uncertainty distribution at four different imaging depths in the fly brain cleared with ScaleView-A2. Scale bar = 20 μm in (**b–d**), 1 μm in (**e**). Source data are provided as a Source Data file

Two-dimensional (2D) localization density (number of localization events/area of the structure) increases exponentially with the number of frames. As the number of sampling frames increased, growth of localization density eventually slowed because of photo-bleaching and a depletion of dye molecules (Fig. 3d). The theoretical resolution limit of localization density was estimated by localization precision and the Nyquist resolution[26]. Resolution power increased rapidly in the first 400 frames (Fig. 3e) but then slowed down and converged at about 60 nm. This represents a lower bound at which the finest structures can be resolved with the current imaging parameters (Fig. 3e). Similarly, the Fourier ring correlation (FRC) value, which reflects the finest structure can be distinguished in the image, decreases with the frame (Fig. 3f)[27,28]. If an infinite photon budget existed and no photo-bleaching occurred, the localization density would increase monotonically with the number of sampling frames. Such modeling indicates that the theoretical resolution may reach 20 nm when 15,000 frames per layer are used for image reconstruction, which represents a 25-fold increase in the acquisition time used for this study (Fig. 3g).

With LLM-CT, the super-resolved image quality of olfactory projection neurons was maintained throughout the entire adult brain (Fig. 4a, b, Supplementary Movie 3). Notably, the photo-electric sensors in conventional fluorescence imaging have a limited dynamic range for detection sensitivity. Thus, simultaneously capturing structures with both strong and weak fluorescence signals in a single image is difficult (Fig. 4c). In localization microscopy, the individual molecules are reconstructed separately based on the histogram of localization events, and consequently the imaging is less sensitive to intensity differences in a single image. When

imaging densely bundled neural fibers connecting the brain with the body, for instance, LLM-CT captured significantly more fibers than LM-CT (Fig. 4d), and individual brain-ascending/descending fibers could be manually segmented (Fig. 4e). LLM-CT also enabled the 3D visualization of individual synaptic proteins. For example, we were able to image the Down syndrome cell adhesion molecules (Dscam) within a single spine-like protrusion on a dendrite of the giant fiber neuron (Fig. 4f, g).

**Visualizing fine neurites in the whole adult brain**. To apply our LLM-CT system to intact adult brain, HMSiR was immunostained to the green fluorescent protein (GFP) labeled dopaminergic neurons with a *TH-Gal4*[29] using anti-GFP antibody (Methods), and mapped the distribution and wiring patterns of all *TH-Gal4* neurons throughout the adult brain (Fig. 5a). Typically, more than 100 million HMSiR blinking signal were detected within each of the four sub-volumes used to reconstruct super-resolution images of the entire brain during a <1-day acquisition time (Methods). To analyze the resulting large dataset of raw blinking images for whole-brain reconstruction, we created a parallel computing pipeline process based on ThunderSTORM[22] on a Lustre-backed Torque cluster (Supplementary Fig. 7). This large-volume super-resolution protein mapping enabled simultaneous visualization of the putative dopaminergic neurons labeled with strong GFP signal in the central brain, as well as the fine neurites labeled with weak GFP signal in the optic lobe (Fig. 5b, c, e). A majority of MB-innervated dopamine neurons direct their axons into the vertical lobe[30]. We calculated the innervation density of *TH-Gal4*-labeled MB neurons in each sector of the vertical lobe and determined that the middle part of the

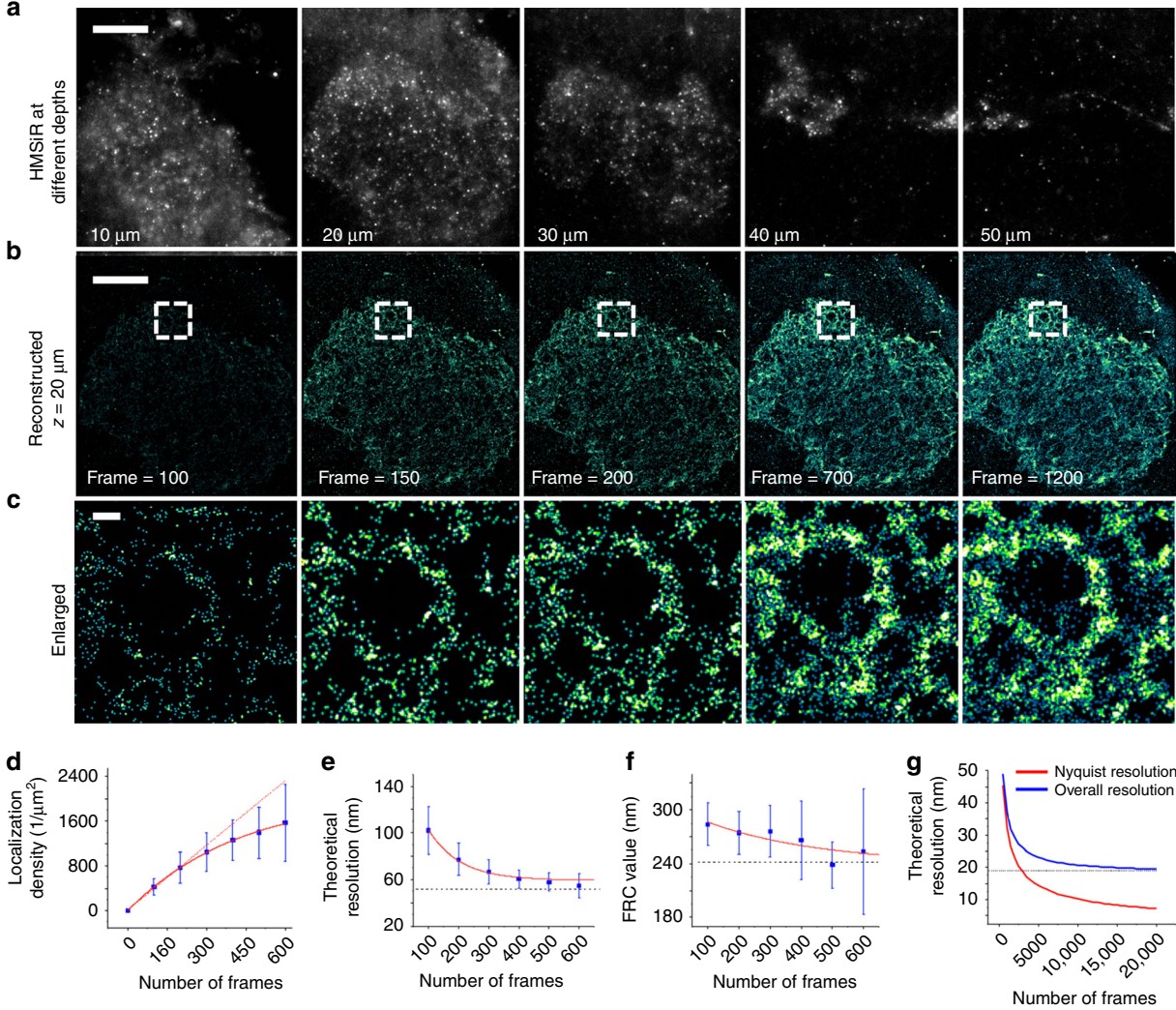

**Fig. 3** Super-resolved three-dimensional neuron images in *D. melanogaster*. **a** At different imaging depth, the blinking signals of HMSiR were imaged through the volumetric scan of lightsheet. **b** Super-resolution image of neuron at specific imaging depth were reconstructed with different length of time-lapse images, which corresponds to different localization density. **c** An enlarged region of interest (white box) in **b** is shown to compare the increased resolution resulted from the higher localization density. **d** Localization density increases exponentially with respect to the number of frames used in reconstruction. As the number of frames reaches 600, the growth of the localization density reaches a plateau due to the depletion of fluorophore and photobleaching. A linear increase was noted in the localization density (red dotted line), assuming unlimited photons budget is drawn for comparison. **e** The theoretical resolution increased rapidly at the beginning. No significant increase was noted with the number of frames exceeding 400 and the lateral resolution at about 60 nm. **f** Fourier ring correlation (FRC) analysis of the localized molecules expressed in a fly's olfactory projection neuron. The decrease in FRC value at 1/7th cut-off frequency is shown with respect to the number of frames. **g** The estimation of Nyquist and overall resolution assuming a linear increase of localization density (no photobleaching and the depletion of fluorophores present). Scale bar = 20 μm in (**a**, **b**), 1 μm in (**c**). Each value represents mean ± S.D. (*n* = 4). Source data are provided as a Source Data file

vertical lobe (sectors α2, α′2) had the highest innervation density (Fig. 5d).

By applying the defocusing PSF model during post-processing of the data (Supplementary Note 1; Supplementary Figs. 8 and 9), digital segmentation of individual neurons, with nominal axial overlap, was achieved in a whole-brain 3D image (Fig. 5e, f). Serial optical slices and 3D navigation demonstrated extensive yet separable neurites (Supplementary Movie 4).

**Visualizing protein spatial distribution within a single neuron.**
Expression level and spatial distribution of proteins are essential to the function of a cell. Scientists historically quantify such endogenous proteins with immunostaining and visualize the label intensity at the cell body with confocal or wide-field microscopy to represent the expression level (Fig. 6a). Like many proteins, the

vesicular monoamine transporter (VMAT) needs to be transported and/or locally synthesized at neuronal terminals to be functional and might be regulated locally by modulatory signals[31], resulting in variable expression within a given neuron[32]. We explored this idea by visualizing VMAT molecules in a specific MB local neuron–the dorsal paired medial (DPM) neuron, located in the central brain, and calculated the density of VMAT protein along the different sub-regions in the MB.

We labeled VMAT with HMSiR and used our LLM-CT system to reconstruct VAMT locations. VMAT molecules along the MB lobes were classified into two groups: DPM positive (DPM+) and DPM negative (DPM-), based on the 3D digital intersection between VMAT and DPM (Fig. 6b, e, h; Supplementary Movie 5). Importantly, this classification could be accomplished only by using our LLM-CT method (Fig. 6b) and not by traditional Airyscan

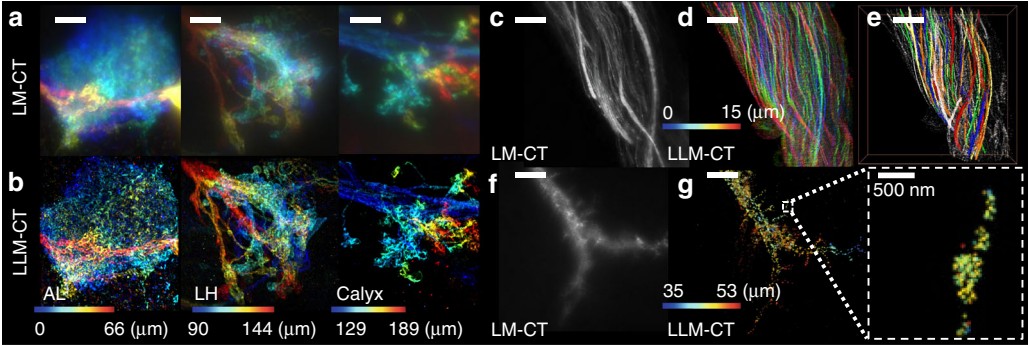

**Fig. 4** Super-resolution images in *D. melanogaster* brain with LLM-CT. Comparison between LLM-CT and LM-CT. **a**, **b** Images of *MZ19-Gal4*-labeled olfactory projection neurons in the antennal lobe (0–66 μm), lateral horn (90–144 μm), and calyx (129–189 μm) taken with LM-CT (**a**) or LLM-CT (**b**). **c**–**e** Images of *Fru-Gal4*-labeled fly neck neurons taken with LM-CT (**c**) or LLM-CT (**d**). **e** More than 30 neurons were segmented manually across a 15-μm imaging depth. **f**, **g** The dendritic arbors of a giant-fibre neuron expressing *UAS-Dscam::GFP* imaged with LM-CT (**f**) or LLM-CT (**g**). 3D images were color-depth coded in each figure. Scale bars = 10 μm, unless otherwise indicated

**Fig. 5** LLM-CT imaging of all dopaminergic neurons in *D. melanogaster* brain. **a** A whole-brain super-resolution mapping of all dopaminergic neurons labeled by HMSiR via anti-green fluorescent protein immunostaining in *TH-Gal4 > UAS-GCaMP6f* flies. **b** Maximum intensity projection of several optical slices showing dopaminergic innervations in the ellipsoid body (EB) and fan-shaped body (FB). **c** Dopaminergic innervations in the mushroom body (MB). Inset: magnified view of the optical slices at the α2α′2 sector of the MB vertical lobe. The colors indicate different depths beneath the surface. Scale bars = 20 μm, unless otherwise indicated. **d** Super-resolved dopamine neuron fiber density distribution in each sector of the MB. **e** Volume rendering of whole-brain dopaminergic neurons. Zoomed-in images show distinguishable interweaved neurites. **f** Digital segmentation of local neurons in the medulla. Inset: enlarged green neuron. The experimental flies carried *TH-Gal4*; *UAS-GCaMP6f* transgenes

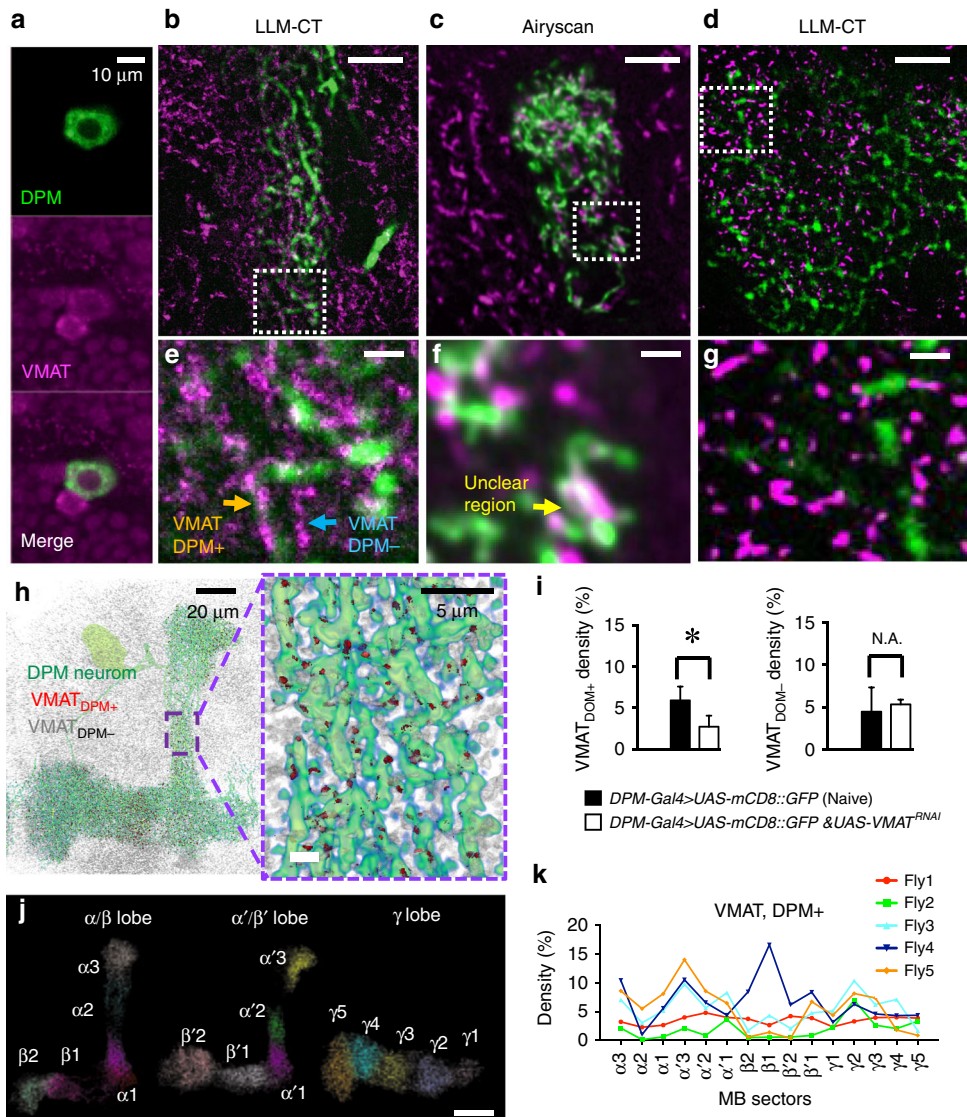

**Fig. 6** Co-localization between VMAT molecules and DPM neurites in the whole MB. **a** VMAT signal in the soma taken by Zeiss 880 confocal microscopy. **b** LLM-CT showed distinguishable signal of anti-VMAT immuno-positive molecules (magenta) were within or outside the DPM neurites (green). **c** As a comparison, Airyscan super-resolution confocal microscopy provides relatively poorer resolution. The ambiguous overlapping signals prevent the high accuracy co-localization analysis. **d** Levels of anti-VMAT immuno-positive molecules decreased within but not outside the DPM neuron in the flies carrying *UAS-mCD8::GFP;VT64246-Gal4/UAS-VMAT-RNAi* transgenes. **e–g** Magnified views of the images in the white boxes of (**b–d**). VMAT was immunolabeled with Alexa Fluor 635 (**a, c, f**) or HMSiR (**b, d, e, g**). **h** Visualization of VMAT molecules. Left, VMAT proteins (gray) distributed within (red) or outside (gray) the DPM neurons (green). Right, an enlarged view. **i** *VMAT^RNAi*-mediated changes in VMAT density in the DPM + neurites (left) but not in DPM- regions (right) within the MB. Each value represents mean ± S.E.M. ($n = 3$). *$P < 0.05$. **j** A representative volume image showing the 3D distribution of VMAT in the DPM neurites innervating MB lobes: α/β (α1, α2, α3, β1, β2), α′/β′ (α′1, α′2, α′3, β′1, β′2) and γ (γ1−γ5). Color code indicates VMAT$_{DPM+}$ signals in different MB sectors. **k** Comparison of VMAT distribution in the different MB sectors within the DPM neurites across 5 flies. Scale bar = 5 μm in (**b–d**), 1 μm in (**e–g**) and 20 μm in (**j**) unless otherwise indicated. Source data are provided as a Source Data file

super-resolution confocal microscopy (Fig. 6c, f). To determine the imaging number of frames needed to map VMAT proteins (Supplementary Fig. 10; Supplementary Note 2), we determined theoretical resolution and localization density as discussed in Fig. 3. The reliability of DPM+ VMAT localization was validated by targeting *VMAT^RNAi* expression to DPM neurons, which reduced the total number of VMAT molecules in DPM+ neurites but not in DPM- regions of MB lobes (Fig. 6d, g, i; Methods).

DPM neurites completely engulf both the vertical and horizontal lobes (axonal outputs) of the MBs[33] but activity in the vertical lobes appears to sub serve behavioral responses after a few sessions of olfactory classical conditioning[34]. Furthermore, MB output neurons exhibit learning-gene dependent tuning plasticity[35] and synthesize

new proteins during long-term memory formation[36,37]. We hypothesized that VMAT in a DPM neuron might be distributed differentially to modulate functional differences among MB sectors. Accordingly, we manually segmented DPM neurites into 15 MB sectors: α1, α2, α3, α′1, α′2, α′3, β1, β2, β′1, β′2, and γ1-γ5, based on the pattern of input innervations from dopaminergic neurons (Supplementary Fig. 11; Methods). With quantitative single-molecule localization by LLM-CT, we found that VMAT proteins distribute unevenly within a DPM neuron and their density in each MB sector is highly variable among different flies (Fig. 6j, k). The plasticity of VMAT distribution we observe here may account for the translation of transient MB activity into long-term memory coding at the MB outputs neurons.

Recently, a combination of expansion microscopy and lattice lightsheet microscopy (ExLLSM) has achieved also nanometric resolution in the adult fly brain[38]. For direct comparison between LLM-CT and ExLLSM, we replicated the optical setup in ExLLSM[38] and imaged VMAT expression in the DPM neuron within a 4x expanded fly brain with lattice lightsheet microscopy[13] and compare the result with our data in Fig. 6 (Supplementary Fig. 12). While LLM-CT and ExLLSM both produced high quality images, each has its advantage and limitation (Supplementary Table 2). Though acquisition time only a few more hours than LLM-CT, ExLLSM required to image ~25,000 sub-volumes to cover the expanded fly brain. To record thousands of sub-volumes and process multiterabyte data, customized algorithms to minimize discontinuities between boundaries are required in the lattice lightsheet system[38]. In contrast, LLM-CT imaged the entire brain with only 4 sub-volumes and post-processing using Thunder-STROM could reconstruct the entire brain within a day. For quantitative analysis of protein distribution, LLM-CT using standard immunostaining protocol is preferable (Fig. 6f, h). In ExLLSM, at equivalent high resolution (see Supplementary Table 2), the quality of fluorescent signal in expanded hydrogel is too capricious for quantitative analysis of protein molecules (Supplementary Fig. 12h, i). Finally, LLM-CT exhibits better single-molecule sensitivity than ExLLSM since the latter method suffers from poor detection of signals with a wide intensity distribution (also demonstrated in Fig. 4c).

## Discussion

By combining lightsheet, tissue clearing and spontaneously blinking fluorophores, our LLM-CT method offers a powerful imaging tool for resolving nanoscale 3D structures in an intact adult fly brain. With high-speed localization of single molecules and high-dynamic-range detection of fluorescence, LLM-CT enables high quality images for quantitative investigation of most, if not all, biological processes within or between subjects in details (Methods and Supplementary Tables 1 and 3).

A caveat of using Bessel beam to cover a large sample is the unwanted background induced by the concentric sidelobe. To minimize the deterioration of signal contrast, we used a weak astigmatism and excluded signals with asymmetric PSF during post-acquisition data processing[5] (Figs. 5 and 6; Supplementary Note 1; Supplementary Figs. 8 and 9). Alternatively, we can use two-photon Bessel beam to directly reduce the background[39] in the future (see also Supplementary Figs. 13 and 14).

Another way to improve resolution is using a clearing reagent with high refractive index. With such, the overlap of signals should be reduced and an increase in localization density is expected (Supplementary Figs. 9 and 10). Several clearing reagents with high refractive index can effectively improve tissue transparency, such as FocusClear[24] (used in CLARITY[40]), uDISCO[41], SeeDB2[15], CUBIC[42] and the recently developed FlyClear[43]. Notably, the properties of HMSiR, such as photons per blink and the population of open-forms, are largely affected by the tissue-clearing environment. For this study, we identified ScaleView-A2 (pH 7.4) as the only commercially ready clearing reagent compatible with HMSiR. Going forward, we will need new blinking fluorophores with properties matching diverse tissue clearing reagents and exhibited different emission wavelengths for improved resolution and multi-color imaging, respectively. We need also new algorithms and processing pipelines to improve further the throughput for ever-larger datasets (Supplementary Table 3).

In conclusion, our LLM-CT method provides neurobiologists with a new tool to study nanoscale structures in an intact adult fly brain with less sample perturbation. The high throughput of this process allows for more efficient and reliable statistical analyses

over a large number of samples. Our method can be applied to biological studies investigating 3D subcellular morphologies and protein densities in mm-scale samples. Applying this imaging technique to larger tissue, such as whole mouse brains, may become possible in the future.

## Methods

**Microscope optics and image acquisition.** The schematic of the optical system is shown in Fig. 1a. The beam from a laser combiner equipped with 488 nm (Coherent OBIS 488 nm LS 150 mW), 561 nm (Coherent OBIS 561 nm LS 150 mW), and 637 nm (Coherent OBIS 637 nm LX 140 mW) lasers is expanded to $1/e^2$ diameter of 3 mm by two lenses (7.5 mm FL/5 mm dia (L1, L3, and L5), Thorlabs, AC050-008-A-ML; 30 mm FL/12.5 mm dia (L2, L4, and L6) Thorlabs, AC127-030-A), and each laser is combined with a mirror (Thorlabs, BB1-E02-10 - Ø1" Broadband Dielectric Mirror, 400–750 nm) and two dielectric filters (Semrock, Dichroic Filter (DF): DF1 = LM01-503-25; DF2 = Di03-R561-t1-25 × 36). The expanded laser beam uses an acousto-optic tunable filter (AOTF; AOTFnC-400.650-TN, AA Quanta Tech, Optoelectronic) to control the exposure time and wavelength selection. The laser beam passing through the AOTF is expanded to a $1/e^2$ diameter of 5 mm using a beam expander (Thorlabs, AC254-100-A (L7) and AC254-150-A (L8) Ø1" Achromat, 400–750 nm) to ensure an even Gaussian intensity distribution over the annular ring pattern on the customized aluminum coating[11] (thickness of 1,500 angstroms) as shown in Supplementary Fig. 2a. In the setup of using an axicon for generating a Bessel beam, an additional laser expander (Thorlabs, AC254-50-A (L9) and AC254-200-A (L10) Ø1" Achromat, 400–750 nm) is used to expand the $1/e^2$ diameter of the beam to 20 mm. The axicon (Thorlabs, AX251-A) is placed with an achromatic lens (Thorlabs, AC254-200-A (L11)) to focus the annular pattern onto the quartz mask. A relay lens pair (Thorlabs, AC254-125-A (L12), AC254-100-A (L13), Ø1" Achromat, 400–750 nm) was used to conjugate the ring pattern, created on the mask, to a set of galvanometer scanners (6215H, Cambridge Technology) which are composed of a pair of achromatic lenses (Thorlabs, AC254-100-A (L14 and L15), Ø1" Achromat, 400–750 nm), aligned in a $4f$ arrangement. After passing through the scanning mirror set, the ring pattern is magnified through a relay lens (Thorlabs, AC254-125-A (L16) and AC254-400-A (L17) Ø1" Achromat, 400–750 nm) and conjugated to the back focal plane of the excitation objective (Special Optics, 0.66 NA, 3.74 mm WD, customized objective, N.A. = 0.5, working distance = 12.8 mm; NARLabs, TIRI, Taiwan; for imaging in ScaleView-A2). The annular pattern is projected to the rear focal plane of the excitation objective and forms a self-reconstructive Bessel beam by optical interference. The energy of the beam is confined within the illumination plane while maintaining a sufficiently long propagation length. By positioning the appropriate annulus on the mask, the effective NA was chosen to create a Bessel beam with an axial full-width-at-half-maximum (FWHM) length comparable to the thickness of the specimen. In the inset of Supplementary Fig 2b, a water dipping objective lens (Nikon, CFI Apo LWD 25XW, 1.1 NA, 2 mm WD for imaging in PBS, Olympus, XLPLN25XSVMP2, 25X, 1.05 NA for imaging in ScaleView-A2), orthogonal to the illumination plane and mounted on a piezo scanner (Physik Instrumente, P-725.4CD PIFOC), used to collect the fluorescence signal, which then passes an emission filter (Semrock Filter: LP02-638RU) onto a scientific complementary metal-oxide-semiconductor (sCMOS) camera (Hamamatsu, Orca Flash 4.0 v3 sCOMS) by a 500 mm tube lens (Edmund 49-290, 500 mm FL/50 mm diameter; Tube Lens). A weak astigmatism, used for sub-diffraction imaging in the axial direction, was introduced by inserting a pair of cylindrical lenses (Thorlabs, LK1002RM-A, LJ1516RM-A) between the tube lens and camera.

Although it is reported that HMSiR probe can be excited with very low power density (40 W/cm²), we found it reasonable to increase the power density to 0.1 kW/cm² to increase the photon count per localization event to reduce the localization uncertainty[17]. The power density is estimated by dividing the power measured at the back aperture of the excitation objective to the area scanned by the Bessel beam of specified length. In the optical setup using an annular mask (outer NA = 0.26, inner NA = 0.185 formed in the excitation objective), a Bessel beam with an axial FWHM of 50 μm is generated (Supplementary Figs. 2a, 3g, h. The maximum power measured at the back aperture of the excitation objective is merely 3 mW due to losses within the system and requiring a power density of 0.12 kW/cm² to cover a subarea along the projection neuron (2500 μm²). Although this setup is flexible to vary the beam length using different annular masks, the power loss caused by the mask hinders its applicability to a larger field of view. With the axicon lens combined with an achromatic lens to concentrate the laser energy into a ring pattern, we were able to focus the laser energy on the annular ring mask. The unwanted components were sequentially filtered using an annular ring mask (outer NA = 0.187, inner NA = 0.174 formed in the excitation objective) (Fig. 1, Supplementary Figs. 1 and 3). The length of the lightsheet generated by the axicon lens can be extended over 200 μm to cover the entire cross-section of the fruit fly brain. The power measured at the back aperture is increased to 20 mW, and the power density can thus be maintained at about 0.10 kW/cm² while extending the image area to 20,000 μm² (Fig. 2a).

**Simulation of lens-axicon combination for Bessel beam generation.** The field distribution of the Gaussian beam E($x$, $z$) at the input side of the lens-axicon

combination is similar to $\exp[(ik_0/2R_F - 1/w^2)(x^2 + z^2)]$, where $k_0$ is the propagation constant in the air; $R_F$ is the radius of curvature for the wave front of the beam, and $w$ is the beam width. The FWHM width of the Gaussian beam, $\text{FWHM}_G$, is related to $w$ as $\text{FWHM}_G = \sqrt{2\ln(2)}w$. After passing through the lens-axicon combination, the Gaussian beam gains an extra phase $\Phi = -k_0[(x^2 + z^2)/2F + (n-1)\alpha\sqrt{x^2 + z^2}]$, where $F$ is the focal length of lens; $n$ is the refractive index of axicon, and $\alpha$ is the angle of the axicon. The phase $\Phi$ is responsible for the beam focusing and formation of the ring pattern. As the beam further propagates, its Gaussian profile is gradually converted into a donut shape (ring pattern). The image distance $q$ of the focused ring pattern after the lens-axicon combination can be estimated using the lens law $(F^{-1} = R_F^{-1} + q^{-1})$. The sharpest ring pattern occurs around the image plane at $y = q$ and has an average radius $R_0$ of about $\alpha(n-1)q$. The evolution of the ring pattern near the image plane is simulated based on Fourier optics under paraxial approximation. We used the fast Fourier transformation (FFT) to obtain the transverse beam profiles (on the $xz$ plane) at successive propagation distances, $y$ around $q$, and observed the focusing and defocusing of the ring pattern. The ring pattern obtained at the image plane is then numerically truncated by an indicator function which mimics the ring aperture. This aperture defines the borders of the ring image for the generation of Bessel beam. The truncated pattern is then numerically magnified along the $xz$ plane by 1.6 times to model the expansion of the transverse profile after the $4f$ configuration. This profile is then fed into another series of FFTs which simulates the formation of Bessel beam around the focal point of the final objective lens, and then immersed into a liquid with a slightly higher refractive index (1.38) than that of water (1.33).

**PSF measurement**. As depicted in Supplementary Fig. 3a, the PSF was recorded by imaging fluorescent beads (Thermo Fisher, F8807) immobilized on the surface of a coverslip with Poly-L-Lysine coating (Sigma, P4707). The total scanning range of the lightsheet for PSF recording is 10 μm with a z-step size of 100 nm. The axial profile of the Beesel beam presented in Supplementary Fig. 3b was acquired by imaging the fluorescence of Alexa Fluor 647 (Thermo Fisher, A30679) dissolved in PBS and ScaleView-A2. For the generation of the astigmatism defocusing model used in the super-resolution reconstruction, a cylindrical lens pair, as described in the methods, was inserted between the tube lens and camera. The step size was set as 20 nm for the fitting when using the defocusing model.

**Fly brain mounting**. The samples were loaded on a 5-mm round, glass coverslips (Warner Instruments) in 100 nL of Cell-Tak™ (Corning®). During the image acquisition, the samples and both objectives were immersed in a chamber to maintain optical clarity. All experiments were conducted at room temperature.

**Fly stocks**. Fly stocks were raised on cornmeal food at a temperature of 25 °C and a relative humidity of 70% under a 12-h light/dark cycle. The following fly lines were used in the current study: *Fruitless-Gal4* (66696, Bloomington *Drosophila* Stock Center) was used to label the neck neurons, *MZ19-Gal4* (34497, Bloomington *Drosophila* Stock Center) for the olfactory projection neurons, *12862-Gal4* (111501, DGRC) for the giant fiber neurons, *TH-Gal4* (8848, Bloomington *Drosophila* Stock Center) for the dopaminergic neurons, *VT64246-Gal4* (v204311, VDRC) for the DPM neurons, *UAS-mCD8::GFP* (5137 and 5310, Bloomington *Drosophila* Stock Center) and *UAS-GCaMP6f* (42747, Bloomington *Drosophila* Stock Center) were used as the reporters of Gal4 expression, *UAS-Dscam[1.7]::GFP* (From T. Lee, Howard Hughes Medical Institute, Ashburn, VA) was used to label Dscam in the Gal4-labeled neurons, and *UAS-VMAT RNAi* (v104072 and v4856, VDRC) was used to assess the downregulation of VMAT protein expression in the DPM neurons.

**Immunohistochemistry**. Fly brains were dissected in PBS (pH, 7.2) and immediately transferred to a microwave-safe 24-well plate containing 4% paraformaldehyde prepared in PBS. The plate was placed on a shaker for 25 min. The fixed tissues were permeabilized and blocked in PBS containing 2% Triton X-100 and 10% normal goat serum (NGS; Vector Laboratories, Burlingame, CA) at 4 °C overnight. Immunostaining was sequentially performed in PBS containing 1% Triton X-100 and 0.25% NGS using the following primary antibodies: mouse anti-Discs large antibodies (antibody 4F3; 1:20 dilution; Developmental Studies Hybridoma Bank, Iowa City, IA), rabbit anti-GFP antibodies (1:250 dilution; Thermo Fisher Scientific Inc, A11122), rabbit anti-VMAT (1:250 dilution), and secondary antibodies including either biotinylated goat anti-rabbit immunoglobulin G (IgG) (1:250 dilutions; Thermo Fisher Scientific Inc). The biotin-conjugated IgG were detected using Alexa Fluor 635-streptavidin (1:500 dilution; Thermo Fisher Scientific Inc) or HMSiR-streptavidin (1:1000 dilution). Each step was carried out over the course of 2 days, with extensive washes between the steps at room temperature (25 °C). The samples were then transferred to ScaleView-A2 for 2 h before imaging, or to FocusClear/DeepClear (CelExplorer. Labs Co.) immediately before imaging (Supplementary Fig. 15).

**HMSiR conjugates**. For goat anti-rabbit labeling of HMSiR, we incubated 250 μg of goat anti-rabbit F(ab')2 fragment (Jackson Immunoresearch, West Grove, PA) in 0.1 M sodium borate buffer at pH 8.5 with 0.8 μl of 10 mM HMSiR-N-hydroxy succinimide (NHS) (GORYO Chemical, Bunkyo-ku, Tokyo) in dimethyl sulfoxide

(DMSO) at 37 °C for 30 min, after which the sample was incubated at 4 °C overnight. The conjugated antibody was then passed through a Desalt Z-25 column (emp Biotech GmbH, Berlin) to remove the unconjugated HMSiR-NHS, and the medium was replaced with PBS (pH 7.2). To label HMSiR with streptavidin, we incubated 125 μg of streptavidin (Sigma-Aldrich, St. Louis, MO) in 250 μl PBS at pH 7.2 with 0.8 μl of 10 mM HMSiR-NHS in DMSO, after which we utilized the same protocol as that used for antibody labeling. Concentrations were then measured at an absorbance of 280 nm, after which the HMSiR conjugates were stored at 4 °C until use.

**Localization image reconstruction**. Localization images were reconstructed using ThunderSTORM[22], an ImageJ plugin with an in-built macro for batch processing of large data. As schematically depicted in Supplementary Fig. 7, the data were initially transposed to a time-lapse series on a local workstation, after which they were transferred to remote Lustre storage and distributed to a three-node Torque cluster (Intel Xeon X5660 with 48 GB memory each, connected to Lustre storage). After a particle list was generated for each layer, the list was deposited into one of the clusters, after which a single list containing all localization events within the imaging volume was generated. The list was then used to render the reconstructed image for presentation and analysis. ThunderSTORM was used to perform drift correction using fiducial marker tracking or the cross-correlation method. For 3D volume rendering, the image stack was resampled to minimize discontinuities in structural integrity.

**Image rendering, segmentation, and quantification**. Color-coded depth projections of 3D images were created via ImageJ. Blinking signal in Fig. 2d was rendered in Imaris (Bitplane). We further used the Volume Rendering tool in Avizo 9.4 (Thermo Fisher Scientific Inc) for 3D visualization of whole brain dopamine neurons (Fig. 5). For single neuron segmentation, as depicted in Fig. 5, we used the Brush tool in the segmentation mode of Avizo 9.4 (Thermo Fisher Scientific Inc) to manually select the boundary of each single neuron to generate a single neuron boundary mask. We further used the arithmetic tool to intersect the original data with the manually segmented single neuron mask to generate single neuron images with the original gray scale. In Figs. 5d and 6, we used the Material Statistics module in Avizo 9.4 (Thermo Fisher Scientific Inc) to quantify the volume of the MB sectors, DPM neurons, and VMAT protein expression. This module calculates the voxel numbers inside a labeled area, which can be transformed into volumes by multiplying with a known voxel size in each image. We manually segmented the boundaries of the MB sectors using the Lasso tool in segmentation mode. We used the Magic Wand tool to select one seed within the DPM neuron and to determine a reasonable threshold for selecting the connecting voxels. We directly used the Threshold tool for whole-volume VMAT images, setting the lower bound to 78 (8-bit; 0-255).

**Reporting summary**. Further information on research design is available in the Nature Research Reporting Summary linked to this article.

## Data availability
The materials and the data that support the findings of this study are available from the corresponding author upon reasonable request. The source data underlying Figs. 2f, 3d–g, 6f Supplementary Figs. 10a, b are provided as a Source Data file.

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

## Acknowledgements

The control software of the microscope is licensed by Howard Hughes Medical Institute, Janelia Farm Research Campus. We thank Professor Tim Tully for valuable suggestions. We thank the Vienna *Drosophila* Resource Center, Drosophila Genetic Resource Consortium, and Bloomington Stock Center for the fly stocks. We thank David Krantz for sharing the rabbit-anti VMAT antibody. This work was supported by grants from the Ministry of Science and Technology and Ministry of Education of Taiwan for L.A.C. (MOST 106-2321-B-007-008-MY3), A.S.C. (Supported by the Higher Education Sprout Project funded by the Ministry of Science and Technology and Ministry of Education in Taiwan) and B.C.C. (MOST 107-3017-F-007-004, MOST 103-2113-M-001-003-MY2, MOST 108-3114-Y-001-002, MOST 107-0210-01-19-01 and Career Development Award and AS-SUMMIT-108 of the Academia Sinica).

## Author contributions

L.-A.C. and C.-H.L. planned and performed the imaging experiments and image processing. L.-A.C. performed the immunostaining experiments. L.-A.C., C.-H.L., Y.-T.L., and B.-C.C. designed and constructed the LLM-CT. S.-M.Y. produced the HMSiR conjugates. Y.-T.L. designed the automated blinking image processing protocol and software. K.-L.F. and L.-A.C. performed the VMAT experiments. L.-A.C., C.-H.L., and Y.-C.T. performed the image analysis. W.-K C. performed the graphic design and system maintenance. L.A.C. and W.-C.W. performed data visualization. S.-W.C. performed the simulation. P.C., Y.-K.H., T.-K.L., A.-S.C., and B.-C.C. planned and managed the project. L.-A.C., C.-H.L., B.-C.C., and A.-S.C. wrote the paper.

## Competing interests

The authors declare no competing interests.
