## [Peer Review File · Nature Communications]

Reviewers' comments:

Reviewer #1 (Remarks to the Author):

In this manuscript the authors describe and demonstrate an approach for localization-based super-resolution microscopy in cleared tissue using a scanned Bessel beam light sheet. Overall, it is an interesting combination of methods that gives a promising result, however, there are many missing details that should be included and other issues.

Custom microscope systems are now usually described in more detail for this level of journal, including part numbers for all lenses used. For example, the axicon lens, an element described as important for improving the Bessel beam, is only given as (ThorLabs), where many axicons are available now from Thorlabs. Other lenses are not given either in part number or focal length. The authors should include a clear summary of their instrument.

The scanning rate of the swept Bessel beam is not given in the methods. It is also not clear if intensity stated means instantaneous power at the center of the Bessel beam, or time average power over the sweep.

It is not clear what the fundamental advantage of the axicon lens is. For properly implemented mask and axicon, the only difference should be power efficiency.

It is not clear what exactly is being imaged in SI Fig. 1. What dye, what sample? At what depth?

SI table 1 should include all figures for which that dataset applies. For example, which data set does SI Fig 3, 4, and 6 belong?

Main text says Figure 2 has data collected using astigmatism/cylindrical lens. Figure 2 caption doesn't mention this. I guess text is wrong.

One of the main points is that clearing the tissue will allow good single molecule imaging throughout the sample. But there must be some degradation of imaging, e.g. localization precision, with depth through the sample to the imaging objective.

Overall, the paper could greatly benefit from a more straightforward presentation. Authors should just clearly describe what the best implementation is and how good it is in various aspects. For example, if Bessel beam with axicon is better, just use that and leave the comparisons to old methods to supplement. Same with cylindrical lens and astigmatism. I did like the analysis of what can be done with the limitations of the dye and FRC.

After a reorganization, authors should make sure methods are complete and detailed.

Reviewer #2 (Remarks to the Author):

In this study, the authors combined ScaleView-A2 tissue clearing, blinking dyes and axicon improved light sheet microscopy to resolve Dopaminergic neurons in a dissected entire *D. melanogaster* brain with subcellular resolution and fast acquisition speed.

Minor comments:

-I would write *D. melanogaster* instead of *Drosophila*. You are working with a specific species not with an entire family.

-A clear structure is missing (Abstract, Introduction, Results and Discussion). Please organise your manuscript according to the formatting standards of the journal you are submitting

-authors claim that they are able to produce a light sheet 0.5 μm thick and 200 μm wide. However

0.5 μm is only the thickness of the central peak of the Bessel beam whereas the surrounding rings make the axial resolution much worse. Thus the authors should provide a clear optical characterisation of their light sheet. They should also show the z axis resolution in their 3D reconstructions and rotate their reconstructions in all axes. For what they want to do I expect they have to resort to a lattice light sheet microscope.

-The authors point out the need for high refractive index matching media with a certain pH value. However, to my knowledge there are a number of solutions with higher refractive index than ScaleView-A2 that can be easily pH adjusted e.g. different Iodide solutions like meglumine diatrizoate in PBS. Further, the authors cited the method SeeDB. I would recommend to cite the follow up method SeeDB2. This method uses RI medium called Histodens which can be pH adjusted and has a higher RI value than ScaleView-A2.

Major comments:

Authors are completely missing to discuss recent publications involving light sheet imaging in combination with *D. melanogaster* clearing e.g. "Cortical column and whole-brain imaging with molecular contrast and nanoscale resolution" doi: 10.1126/science.aau8302 and "High-resolution ultramicroscopy of the developing and adult nervous system in optically cleared *Drosophila melanogaster*" doi.org/10.1038/s41467-018-07192-z.

In one manuscript entire flies are imaged and in the other publication super resolution images of *D. melanogaster* neurons are displayed with better resolution than in this manuscript. I would request a direct comparison of the present manuscript with both of the aforementioned approaches in an additional Supplementary Figure. The novelty and advantages of the manuscript should be pointed out, that would justify a publication in a high impact journal, as Nature Communications.

Reviewer #3 (Remarks to the Author):

The major claim of the paper is the integration of single molecule localization techniques, tissue clearing, and Bessel beam light sheet microscopy for super resolution imaging of a large intact structure, the fly brain. This combination is novel and will be of interest to both the microscopy and biology communities, but requires further justification and explanation of techniques used.

Within the methods section, the authors acknowledge the relationship between the illumination geometry, axial extent of the Bessel beam, and residual side lobe energy - these are essential considerations in light sheet microscopy, but especially in its application to single molecule localization, which requires high intensity illumination and high signal contrast. Their comparison of various illumination geometries, either experimental or theoretical, would be useful to their argument that their chosen imaging parameters allow for high enough light intensity for HMSiR blinking and low concentric sidelobe background. One related claim that is integral to their ability to image an entire fly brain in a reasonable amount of time is the extension of the Bessel beam light sheet from 50 to 200 microns. This is done by creating the Bessel beam with an axicon lens in contrast to annular ring mask; however, it is unclear if their claim is meant to address the fundamental issue of high energy side lobes present in this scheme, except to say that "the unwanted components were filtered using an annular ring mask" (Methods, PG 15 line 2-3). To this reader, the diagram in Supplementary Figure 1 does not explain this claim. They then say in their discussion that "unwanted concentric sidelobe background may deteriorate the signal contrast," and offer as an alternative two-photon excitation. The authors should justify their use of the described illumination scheme, in contrast to structured illumination or two-photon Bessel beam light sheet, and clarify their claims on the best methodology to achieve high enough contrast for single molecule localization at the resolutions described (can they still demonstrate their axial resolution of 500 nm with these high intensity concentric side lobes?). Some finer notes: they should be more clear in figure legends which illumination scheme was used (axicon or ring mask), and should not switch between ID/OD/NA in manuscript text and figures. Their claim "to improve

the resolution in the axial direction, optical astigmatism has been used (shown in Fig. 2)” does not seem to refer to the appropriate figure.

Similarly essential to their ability to image an entire fly brain is the use of ScaleView-A2 to clear the tissue, which was compatible with HMSiR, a blinking fluorophore that has been well described for single molecule localization. It is interesting and impactful to demonstrate suitable combinations of clearing solutions and blinking fluorophores. They should take care with their claims that super resolution has not been used within large, cleared tissues, and should acknowledge the contributions of Long et al. 2017 (doi:10.1038/nmeth.4309), who used Bessel beam SIM to localize single RNA molecules. Their use of BB-SIM on cleared tissue should be acknowledged, perhaps especially because in theory it should provide more optimal signal contrast.

While the authors have imaged an impressive and perhaps unprecedented level of detail across large spatial scales, there is little characterization of the biological structures that they have imaged and supposedly resolved. For such a large and heroic dataset and subsequent analysis, the reconstruction of 30 neurons (“The improvement in LLM-CT allows more than 30 neurons were traced and segmented”) is underwhelming. Further, it was unclear to me how neurite reconstruction was performed, post processing with ThunderSTORM, so some citation or description of appropriate algorithms might be missing.

I believe this manuscript has a value and novelty appropriate for Nature Communications, and with increased justification and clarity of the illumination parameters and some greater quantification of structural resolution within their biological samples, this manuscript could be accepted.

Erin Diel, PhD
Imaging Specialist, Harvard Center for Biological Imaging

We appreciate the positive recognitions and valuable suggestions from the reviewers. Following their suggestions, we have made necessary revisions, as well as additional characterization of our image system and super-resolved neurons. Attached please find our point-to-point responses, as well as the revised manuscript. To assist reading, the reviewers' comments are marked in black color, our responses are in blue, and the revised sentences in the manuscript are in green.

Due to the restructure and more detailed system description of our manuscript based on reviewer's suggestions, we listed major changes in bellow instead of mark all changes in the manuscript.

1. We restructured the manuscript into introduction, results and discussion.
2. We added several paragraphs in the manuscript, including main text and methods, to describe the imaging system in detail, and move the resolution analysis from Methods to the main text (Fig. 3).
3. We added a new movie (Movie 3) to demonstrate the 3D super-resolution image of the axon terminal arbors of olfactory projection neurons in lateral horn.
4. We split original Fig. 2 into Fig. 4 and 5 for a more detailed description of the results.
5. We added Fig. 6 for a new biological application of our system.
6. We added a direct comparison between ExLLSM and our LLM-CT based on reviewer 2's request.

Reviewer #1 (Remarks to the Author):

In this manuscript, the authors describe and demonstrate an approach for localization-based super-resolution microscopy in cleared tissue using a scanned Bessel beam light sheet. Overall, it is an interesting combination of methods that gives a promising result, however, there are many missing details that should be included and other issues.

Custom microscope systems are now usually described in more detail for this level of journal, including part numbers for all lenses used. For example, the axicon lens, an element described as important for improving the Bessel beam, is only given as (ThorLabs), where many axicons are available now from Thorlabs. Other lenses are not given either in part number or focal length. The authors should include a clear summary of their instrument.

Response: Thanks for the suggestion. We have indicated the part number of all lenses used in the study and updated in the "Methods" section as follow (Page 13-14): "The schematic of the optical system is shown in Figure 1a. The beam from a laser combiner equipped with 488 nm (Coherent OBIS 488 nm LS 150 mW), 561 nm (Coherent OBIS 561 nm LS 150 mW), and 637 nm (Coherent OBIS 637 nm LX 140 mW) lasers is expanded to a 1/e² diameter of 3 mm by two lenses (7.5 mm FL/5 mm dia (L1, L3, and L5), Thorlabs, AC050-008-A-ML, 30 mm FL/12.5 mm dia Thorlabs, AC127-030-A (L2, L4, and L6)) and combine each lasers with a mirror (Thorlabs, BB1-E02-10 - Ø1" Broadband Dielectric Mirror, 400 - 750 nm) and two dielectric filter (Semrock, Dichroic Filter (DF): DF1= LM01-503-25; DF2= Di03-R561-t1-25x36)..... A weak astigmatism used for sub-diffraction imaging in axial direction was introduced by inserting a pair of cylindrical lenses (Thorlabs, LK1002RM-A, LJ1516RM-A) between the tube lens and camera."

The scanning rate of the swept Bessel beam is not given in the methods. It is also not clear if intensity stated means instantaneous power at the center of the Bessel beam, or time average power over the sweep.

Response: In response to reviewer's question, the intensity we stated here is the time-averaged power over the sweep. The excitation power density is calculated with the method as previously described (Legant, et al, Nat. Methods 13 359-365, 2016, DOI: 10.1038/nmeth.3797). It is estimated by dividing the power measured at the back pupil of the excitation objective to the area covered by lightsheet ((beam length) x (scanning range)). Our Bessel lightsheet microscope (Supplementary Fig. 2) uses a galvanometer to sweep the Bessel beam in the x direction across the plane of focus of the detection objective to create a time-averaged lightsheet. The scanning rate is dependent on the exposure time and the number of steps used in galvanometer scanning. We show the timing diagrams showing waveforms for the swept sheet mode in Supplementary Fig. 2c.

In Supplementary Table 1, we added exposure time information for different data sets and the scanning areas are listed as well. To clarify this point, the following sentence was added in the revised manuscript main text (page 14). "The power density is estimated by dividing the power measured at the back aperture of the excitation objective to the area scanned by the Bessel beam of the specified length."

It is not clear what the fundamental advantage of the axicon lens is. For properly implemented mask and axicon, the only difference should be power efficiency.

Response: To strengthen the advantages of using axicon lens, we added our simulation results about lens-axicon combination to create a variety of Bessel beam lengths by changing the beam width of illuminated Gaussian beam (See the revised Fig. 1b, 1c and Supplementary Fig. 1). As reviewer mentioned, the main reason we used the axicon to generate Bessel is to increase the power efficiency. Importantly, a thinner annular mask generates longer light beam that extends the lightsheet scanning area (Gao, L. *et al. Nat Protoc* 9, 1083-1101 (2014), Chen, B.-C. *et al. Science* 346, 1257998 (2014). However, the drawback of this approach (annular mask) is more laser power would be blocked from passing through the mask.

In order to maintain the laser power density while using longer lightsheet for large sample, we use axicon to reshape the illumination profile to fit the annular mask. By adapting this modification, the laser power passing through the mask can be increased. We can thus make use of the improved power throughput to increase the length of Bessel beam and be able to extend the field of view by using the entire camera chip of the sCMOS camera.

To make this concept clearly explained in the manuscript, we added the following sentences in the main text on page 5: "Instead of using a spatial light modulator to generate lightsheet in LLSM, the lightsheet for localization application in our scheme was generated from a scanning Bessel beam created by a lens-axicon doublet, illuminated by a Gaussian laser beam¹⁵ (Fig. 1b). Illuminated by varied Gaussian beam sizes, different ring-shaped focal patterns were formed, which were tuned for the modulating the lightsheet length (Supplementary Fig. 1)."

It is not clear what exactly is being imaged in SI Fig. 1. What dye, what sample? At what depth?

Response: We added the necessary information in the revised Supplementary Fig. 3 (original SI Fig. 1) and provided additional details on the acquisition of point spread function and the axial profile of Bessel beam on page 16: “As depicted in Supplementary Fig. 3a, the PSF was recorded by imaging fluorescent beads (Thermo Fisher, F8807) immobilized on the surface of a coverslip with Poly-L-Lysine coating (Sigma, P4707). The total scanning range of the lightsheet for PSF recording is 10 μm with a z-step size of 100 nm. The axial profile of the Bessel beam presented in Supplementary Fig. 3b was acquired by imaging the fluorescence of Alexa Fluor 647 (Thermo Fisher, A30679) dissolved in PBS and ScaleView-A2. For the generation of the astigmatism defocusing model used in the super-resolution reconstruction, a cylindrical lens pair, as described in the methods, was inserted between the tube lens and camera. The step size was set as 20 nm for the fitting when using the defocusing model.”

SI table 1 should include all figures for which that dataset applies. For example, which data set does SI Fig 3, 4, and 6 belong?

Response: Thanks for the reviewer’s correction. We added the information in the revised Supplementary Figs. 5, 6, main Fig. 5 (original SI Figs. 3, 4 and 6, respectively) and the Supplementary Table 1.

Main text says Figure 2 has data collected using astigmatism/cylindrical lens. Figure 2 caption doesn’t mention this. I guess text is wrong.

Response: Thanks for the reviewer’s correction. Data presented in original Fig. 2a, 2b, 2c) are indeed obtained without astigmatism/cylindrical lens. Because of smaller field of view ($\sim 50 \mu\text{m}$) (around $\frac{1}{4}$ camera chip size), we used the shorter lightsheet with less background contributed from the sidelobe of Bessel beam. However, when a large field of view ($\sim 200 \mu\text{m}^2$) is needed for imaging the whole fly brain (shown in original Fig. 2d, 2e), a longer Bessel lightsheet is used and significant amount of sidelobe background is inevitable. To compensate the higher background induced by the sidelobe, a weak astigmatism was applied in the detection path to allow the exclusion of the out of focus signals in post-processing. In the revised manuscript, we separated original Fig. 2 into two figures (Fig. 4 and Fig. 5) to make the point clear. Moreover, we used Supplementary Note 1 and Supplementary 8 and 9 to describe the method we used for the background removal.

One of the main points is that clearing the tissue will allow good single molecule imaging throughout the sample. But there must be some degradation of imaging, e.g. localization precision, with depth through the sample to the imaging objective.

Response: Thanks for the reviewer’s comment. To better illustrate the effect of tissue clearing on the single-molecule imaging in thick tissues, we compared image quality of HMSiR blinking signal in the fly brain kept in PBS or ScaleView-A2 medium at the same depth (revised Fig. 2b, 2c). We also showed the improved precision of single-molecule localization in the clarified tissues throughout the entire fly brain (revised Fig. 2d, 2e, 2f). As for the degradation of imaging, we discuss the relation between localization uncertainty and imaging depth in Supplementary Fig. 8 and Supplementary Note 1. In this discussion, we characterized the image degradation and used the post-processing procedure to compensate the degradation.

Overall, the paper could greatly benefit from a more straightforward presentation. Authors should just clearly describe what the best implementation is and how good it is in various aspects. For example, if Bessel beam with axicon is better, just use that and leave the comparisons to old methods to supplement. Same with cylindrical lens and astigmatism. I did like the analysis of what can be done with the limitations of the dye and FRC.

Response: As reviewer suggested, we presented our lens-axicon method for Bessel beam generation in the revised manuscript and leave traditional method (mask) in the supplement information. Furthermore, we added an introduction and explained the advantages of the lens-axicon method in the revised Fig. 1 and Supplementary Fig. 1. For the cylindrical lens and astigmatism as reviewer pointed out, our strategy of using it as a compensation for the aberration is discussed in the supplement, as mentioned in the previous answers.

Thanks to the reviewer's positive feedback about the FRC analysis. We further elaborated the analysis and moved the data to Figure 3 in the main text. We also added more raw images about how the super-resolve 3D neuron image in the antennal lobe of *D. melanogaster* was reconstructed based on the accumulation of localized molecules over time (Fig. 3a, 3b, 3c). The descriptions about the FRC and limitation analysis were added on page 7 in the revised manuscript as "To determine a realistic acquisition time with a reliable statistical basis, we plotted the theoretical resolution with respect to time based ... which is 25-fold increase of the acquisition time used in this study (Fig. 3g)." We also included Supplementary Note 2 to discuss the theoretical resolution and FRC analysis of the dataset used in Fig. 6.

After a reorganization, authors should make sure methods are complete and detailed.

Response: As the reviewers will see, we have reorganized the manuscript and provided completed and detailed methods as suggested (see pages 13-17, Supplementary Note 2").

Reviewer #2 (Remarks to the Author):

In this study, the authors combined ScaleView-A2 tissue clearing, blinking dyes and axicon improved light sheet microscopy to resolve Dopaminergic neurons in a dissected entire *D. melanogaster* brain with subcellular resolution and fast acquisition speed.

Minor comments:

-I would write *D. melanogaster* instead of *Drosophila*. You are working with a specific species not with an entire family.

Response: We have changed *Drosophila* to *D. melanogaster* through the entire manuscript as suggested.

-A clear structure is missing (Abstract, Introduction, Results and Discussion). Please organise your manuscript according to the formatting standards of the journal you are submitting

Response: We have reorganized the manuscript into *Nature Communications* format.

-authors claim that they are able to produce a light sheet 0.5 μm thick and 200 μm wide. However 0.5 μm is only the thickness of the central peak of the Bessel beam whereas the surrounding rings make the axial resolution much worse. Thus, the authors should provide a clear optical characterisation of their light sheet. They should also show the z axis resolution in their 3D reconstructions and rotate their reconstructions in all axes. For what they want to do I expect they have to resort to a lattice light sheet microscope.

Response: Thanks for the reviewer's comment. For a thin (\sim half a micron) and long Bessel beam (\sim 200 μm), the background contributed from the sidelobes of Bessel beam is inevitable, resulting in worse axial resolution. That's the reason we introduce a weak astigmatism in the detection to improve the precision of both lateral and axial position determination. This approach is explained in Supplementary Note 1 and Supplementary Fig. 8 and 9. The characterizations of the axial profile of the Bessel beam and PSF are shown in Supplementary Fig. 3. Moreover, for the astigmatism characterization, we have thoroughly discussed pros and cons between Bessel/lattice light sheet microscope in a recent publication (*Communications Biology* DOI:10.1038/s42003-019-0403-9). We will cite this reference in the revised manuscript (Lu, C.-H. *et al. Communications Biology* 2 (2019)). We also added a Supplementary Movie 3 to show our 3D reconstructions with rotating animation as the reviewer suggested. The main reason that we don't use lattice light sheet microscope (LLSM) as the platform for the single-molecule localization in clarified fly brain is as following: LLSM is the result of coherent interference between Bessel beams, which requires an aberration-free environment such as single cell or multicellular system. In our system, ScaleView-A2 (refractive index= 1.38) treated fly brain as shown in Supplementary Fig. 15 is still translucent, caused by the mismatch of refractive indexes between tissue and surrounding medium. Therefore, lattice light sheet won't form correctly because the delicate structure of the laser wave front is distorted by the residual scattering in the tissue. Therefore, we go with scanning a Bessel beam, which is a self-reconstruction beam and less sensitive to the varied refractive indexes. We have addressed these points in the revised manuscript on page 5: "The Bessel beam has been described as a self-reconstruction light beam that is particularly effective for penetrating a thick specimen^{16,17}".

-The authors point out the need for high refractive index matching media with a certain pH value. However, to my knowledge there are a number of solutions with higher refractive index than ScaleView-A2 that can be easily pH adjusted e.g. different Iodide solutions like meglumine diatrizoate in PBS. Further, the authors cited the method SeeDB. I would recommend to cite the follow up method SeeDB2. This method uses RI medium called Histodens which can be pH adjusted and has a higher RI value than ScaleView-A2.

Response: Thanks for reviewer's suggestion for a higher refractive index medium. Higher refractive index medium will defiantly give us better transparency in large tissue and better z resolution as we showed in Supplementary Fig. 14 and 15. However, due to the restriction of the mechanical design of lightsheet microscope, there is no suitable commercially available detection objective lens. We thus choose ScaleView-A2 since the Olympus objective (XLPLN25XSVM2) which can provide optimized image quality at the refractive index of ScaleView-A2 ($n=1.38$). Moreover, the design of the objective with long working distance allows the use of the customized excitation objective, which must be positioned very close to the detection objective lens.

We are keeping an eye on the new high N.A. objectives designed for higher refractive index medium to further improve the precision of single molecule detection in deep tissue. On the other side, the properties of HMSiR fluorophores (photon number and population) are sensitive to the chemical composition of the medium. In our experience, Scaleview-A2 is the best option that can work with our system. The discussion related to this topic can be found in the revised manuscript on page 12 as “Several clearing reagents with high refractive index can effectively improve tissue transparency, such as FocusClear²⁴ (used in CLARITY⁴⁰), uDISCO⁴¹, SeeDB2¹⁴, CUBIC⁴² and the recently developed FlyClear⁴³. Notably, the properties of HMSiR, such as photons per blink and the population of open-forms, are largely affected by the tissue-clearing environment. For this study, we identified ScaleView-A2 (pH 7.4) as the only commercially ready clearing reagent compatible with HMSiR. Going forward, we will need new blinking fluorophores with properties matching diverse tissue clearing reagents and exhibited different emission wavelengths for improved resolution and multi-color imaging, respectively.”

As the reviewer suggested, we cited the method SeeDB2 in the revised manuscript as reference 12.

Major comments:

Authors are completely missing to discuss recent publications involving light sheet imaging in combination with *D. melanogaster* clearing e.g. “Cortical column and whole-brain imaging with molecular contrast and nanoscale resolution” doi: 10.1126/science.aau8302 and “High-resolution ultramicroscopy of the developing and adult nervous system in optically cleared *Drosophila melanogaster*” doi.org/10.1038/s41467-018-07192-z.

In one manuscript entire flies are imaged and in the other publication super resolution images of *D. melanogaster* neurons are displayed with better resolution than in this manuscript. I would request a direct comparison of the present manuscript with both of the aforementioned approaches in an additional Supplementary Figure. The novelty and advantages of the manuscript should be pointed out, that would justify a publication in a high impact journal, as Nature Communications.

Response:

In responses to reviewer’s comments, we first apologized that the reason we didn’t cite *Gao et al.* Science 2019 (ExLLSM) is because it wasn’t formally published before our submission. For the other paper “High-resolution ultramicroscopy of the developing and adult nervous system in optically cleared *Drosophila melanogaster*” we added this reference in the revised manuscript as reference 35 in page 13. In the revision, as reviewer suggested, we have applied ExLLSM to a similar sample for the direct comparison. In Supplementary Fig. 12, the comparison between our technique and ExLLSM by imaging the vesicular monoamine transporter (VMAT) proteins on the DPM neuron in mushroom body (MB) is displayed. Followings are the pros and cons of two methods:

- (1) Sample preparation: In LLM-CT, the fly brain was soaked in the clearing reagent for several hours and can be imaged afterwards. As for ExLLSM, the fly brain sample is expanded by swollen polymer, requiring several steps in days to finish the sample expansion. The protein retention during the expansion process should be taken care, especially for the protein quantitative measurement as demonstrated in our case. Instead, our LLM-CT method could be used to map the distribution of any kind of proteins of interest with standard immunostaining process to label HMSiR molecules

- on targeted protein over thick tissues with nominal sample perturbation (no dehydration or enzyme digestion).
- (2) Spatial resolution: As the data show, although both methods could successfully image the distribution of VMAT proteins, ExLLSM exhibited improved resolution in both channels. On the contrary, in our technique the sub-diffraction limited resolution only applied in one channel. This limitation is resulted from the lack of the second fluorophore but not from the optical design of our system. Since the second fluorophore emits in shorter wavelength has already published (S Uno et al., Chemical Communications, 2018), the application of multi-colors super-resolution imaging by using the proposed technique in this manuscript can be foreseen. We added the following sentences on page 11 in the revised manuscript as “For direct comparison between LLM-CT and ExLLSM, we imaged VMAT expression in the DPM neuron with both methods (**Supplementary Fig. 12**). While LLM-CT and ExLLSM both produced high quality images, each has its advantage and limitation (**Supplementary Table 3**).”
 - (3) Single molecule sensitivity: In wide field microscopy, the dynamic range of the intensity in the image is related to the full well depth of the pixels on the detector. When there are multi fluorophores located in a diffraction limited area, they are not distinguishable and result in a higher signal than a single fluorophore. The higher fluorescence intensity from the crowded fluorophores easily saturates pixel and obstructs the detection of the relative lower fluorescence from single molecule. It limits the possibility of quantitative imaging because the fluorescence intensity can only reflect the protein density distribution in a limited dynamic range. In localization microscopy, even the fluorophores are densely labelling in a diffraction limited area, the stochastic blinking makes it possible to detect them individually in a time-lapse imaging. Besides, the image is reconstructed by counting the number of blinking events, which is not limited by the dynamic range of the detector. As shown in Supplementary Fi. 12b, the VMAT (magenta) distribution mapped by our technique shows more detail. Both the strong and the weak signals are presented in the figure. We believe this feature of our technique not only reveal the structural properties more faithfully, but also provide more robust quantitative information of protein distribution.
 - (4) Acquisition speed: For LLM-CT, our current imaging speed is one fly brain per day (TH-Gal4 labeled dopaminergic neurons). For ExLLSM, according to Gao *et al. Science 2019*, they performed one expanded fly brain ~ 3 days for two-color experiment, which is only slightly longer than our imaging period for single color super-resolution imaging. However, it is worth to know that in ExLLSM, the imaging requires highly automated imaging system to complete 25788 tiled volume scan for each color, and only 4 tiles were needed in LLM-CT to cover the entire fly brain.
 - (5) Data post-processing: For LLM-CT, a reconstructed whole brain super-resolution image was within 100 gigabytes, which makes LLM-CT a suitable tool for qualitative and quantitative comparison within- or between-subject (Method and Supplementary Table 1, 2). For ExLLSM, a 64-fold increase in the volume of a fly brain requires the stitching of 25,000 sub-volume. The image registration and merging of different sub-volumes are required afterwards, which create a significant computation load for data analysis due to the multi-terabyte data size and processing time. The post processed whole fly brain ExLLSM image will still be several terabytes. Practically, our technique is easier to be implemented for biology research community due to the simplicity of data processing and post statistical analysis.

We have added the following information in the revised manuscript on page 11:

“Recently, a combination of expansion microscopy and lattice lightsheet microscopy (ExLLSM) has achieved also nanometric resolution in the adult fly brain³⁸. For direct comparison between LLM-CT and ExLLSM, we imaged VMAT expression in the DPM neuron with both methods (**Supplementary Fig. 12**). While LLM-CT and ExLLSM both produced high quality images, each has its advantage and limitation (**Supplementary Table 3**). Though image acquisition time for the expanded tissue took only a few more hours than LLM-CT, ExLLSM required to process $\sim 25,000$ sub-volumes to reconstruct the entire fly brain. To process large number of sub-volumes and multiterabyte data, customized algorithms to minimize discontinuities between boundaries are required. In contrast, post-imaging data processing for only 4 sub-volumes, LLM-CT could reconstruct the entire brain within a day. Importantly, tissue expansion with enzyme digestion in ExLLSM inevitably causes some protein loss during the lengthy sample preparation, while LLM-CT using standard immunostaining protocols is preferable for quantitative analysis of protein distribution (Fig. 6f, 6h). Finally, LLM-CT images reveal better single-molecule sensitivity than ExLLSM mainly because the latter method suffers from a wide intensity distribution (demonstrated in Fig. 4b).”

Reviewer #3 (Remarks to the Author):

The major claim of the paper is the integration of single molecule localization techniques, tissue clearing, and Bessel beam light sheet microscopy for super resolution imaging of a large intact structure, the fly brain. This combination is novel and will be of interest to both the microscopy and biology communities, but requires further justification and explanation of techniques used.

Response: Thanks for the reviewer’s comments. We have added more justification and explanation of techniques used in the revised manuscript and Supplementary materials. Moreover, we also cited more references to support the present work, especially our recent work published in *Communications Biology*. We provided the reference as attachment in the reply letter to solidify our arguments and address the questions raised here.

Within the methods section, the authors acknowledge the relationship between the illumination geometry, axial extent of the Bessel beam, and residual side lobe energy - these are essential considerations in light sheet microscopy, but especially in its application to single molecule localization, which requires high intensity illumination and high signal contrast. Their comparison of various illumination geometries, either experimental or theoretical, would be useful to their argument that their chosen imaging parameters allow for high enough light intensity for HMSiR blinking and low concentric sidelobe background.

Response: Thanks for the reviewer’s suggestion. In our very recent publication: *Communications Biology*, (DOI:10.1038/s42003-019-0403-9), we have addressed your concerns about various illumination geometries for single-molecule detection by lightsheet. Experimental or theoretical illustration is presented therein (see Supplementary Fig. 8, 9 and Supplementary Note 1).

One related claim that is integral to their ability to image an entire fly brain in a reasonable amount of time is the extension of the Bessel beam light sheet from 50 to 200 microns. This

is done by creating the Bessel beam with an axicon lens in contrast to annular ring mask; however, it is unclear if their claim is meant to address the fundamental issue of high-energy side lobes present in this scheme, except to say that “the unwanted components were filtered using an annular ring mask” (Methods, PG 15 line 2-3). To this reader, the diagram in Supplementary Figure 1 does not explain this claim. They then say in their discussion that “unwanted concentric sidelobe background may deteriorate the signal contrast,” and offer as an alternative two-photon excitation. The authors should justify their use of the described illumination scheme, in contrast to structured illumination or two-photon Bessel beam light sheet, and clarify their claims on the best methodology to achieve high enough contrast for single molecule localization at the resolutions described (can they still demonstrate their axial resolution of 500 nm with these high intensity concentric side lobes?).

Response: Thanks for the reviewer’s comments. The high-energy side lobes from the thin and long Bessel lightsheet will be problematic while we would like to determine the single-molecule 3D positions. To overcome this issue, we introduced the astigmatism in the detection system to improve the spatial resolution, especially for axial directions. We described the illumination scheme in detail in the Supplementary Fig. 1 of the revised manuscript. A characterization of the beam profile is presented in Supplementary Fig. 3. For the effect caused by the side lobes, we made a discussion and presented a strategy to overcome the interference of background signals in Supplementary Note 1 and Supplementary Fig. 8 and 9.

For larger tissue, due to the sample-induced aberration (non-transparent), which deteriorates the single-molecule signal-to-noise ratio makes the localization process more difficult. It will be easier to improve the precision of localization by making the tissue more transparent (see Supplementary Fig. 15) in higher refractive index clearing reagents. However, while we attempt to perform single-molecule localization experiments, we need to use high NA objective (~ 1.1) with long working distance (\sim several mm) and corrected for higher refractive index (~ 1.45 or higher). However, there is no suitable commercially available objective to fit into the design of lightsheet microscope system due to the mechanical and optical restriction. After the tradeoff, we chose ScaleView A2 as the clearing reagent and used Olympus, XLPLN25XSVMP2 (N.A. = 1.05, 2mm WD) as detection objective for the single-molecule detection in the translucent brain. Moreover, longer wavelength (640 nm) laser is used to excite HMSiR molecules, which is less sensitive to the scattering effect of translucent sample. The other important thing is the blinking behavior of single-molecule (HMSiR in our case); the blinking of HMSiR molecule is through the chemical structure transformation. Therefore, the blinking properties will vary in different chemical environments (pH and hydrophobicity). In our experience, Scaleview-A2 is the best option that can work with our system. Based on all these arguments, the present method is optimized for lightsheet localization for clarified tissue to have super-resolved 3D image.

For structure illumination microscope (SIM), it works for aberration-free cellular system and shallow depth because of very high NA (~ 1.27 or higher) objective. Hence, to super-resolve whole fly brain ($\sim 300 \mu\text{m} \times 500 \mu\text{m} \times 200 \mu\text{m}$) with SIM is not feasible. Even if lightsheet-SIM is used to have 3D imaging, the resolution is still not enough to distinguish the DPM and VMAT (in our case) due to the lower NA modulation (~ 0.3) and lower NA detection (~ 1.1) (resolution ~ 200 nm laterally, ~ 500 nm axially in a perfect aligned system). In addition to resolving the fine structures in the sample, we extended the application to map the distribution of interested proteins by localization process.

Responding to these concerns, we addressed the following contents on page 12 in the revised manuscript: “Several clearing reagents with high refractive index can effectively improve tissue transparency, such as FocusClear²⁴ (used in CLARITY⁴⁰), uDISCO⁴¹, SeeDB2¹⁴, CUBIC⁴² and the recently developed FlyClear⁴³. Notably, the properties of HMSiR, such as photons per blink and the population of open-forms, are largely affected by the tissue-clearing environment. For this study, we identified ScaleView-A2 (pH 7.4) as the only commercially ready clearing reagent compatible with HMSiR. Going forward, we will need new blinking fluorophores with properties matching diverse tissue clearing reagents and exhibited different emission wavelengths for improved resolution and multi-color imaging, respectively.”

Some finer notes: they should be clearer in figure legends which illumination scheme was used (axicon or ring mask), and should not switch between ID/OD/NA in manuscript text and figures. Their claim “to improve the resolution in the axial direction, optical astigmatism has been used (shown in Fig. 2)” does not seem to refer to the appropriate figure.

Response: Thanks for the reviewer’s suggestion, to avoid confusion to the readers, we kept the axicon part in the main text and moved the “mask” part to the Supplementary Information as the reviewer 1 suggested. For the ID/OD/NA issue, we made it clear by simulation of lens-axicon combination as shown in revised Fig. 1 and Supplementary Fig. 1, where the radius of the ring formed after axicon in our designed system was indicated to match the ring ID/OD of the lithography aluminum mask. On the other hand, NA was used for the excitation lightsheet formed by our customized objective. We have added more description of the system in the Methods (microscope optics and image acquisition) and specified the imaging condition for each dataset in Supplementary Table 1.

We apologize for the confusion, the question is the same as Reviewer 1, we quoted the answer as following: “original Fig. 2a, 2b and 2c are obtained without astigmatism/cylindrical lens. These are in smaller field of view (~ 50 μm) (around $\frac{1}{4}$ camera chip size), so we could use the shorter lightsheet with less background contributed from the sidelobe of Bessel beam. However, when a large field of view (~ 200 μm^2) is needed to image the fly brain (shown in original Fig. 2d, 2e), a longer Bessel lightsheet is used and significant amount of sidelobe background is inevitable. To compensate the higher background induced by the sidelobe, a weak astigmatism was applied in the detection path to allow the exclusion of the out of focus signals in post-processing. We separated original Fig. 2 into two figures (Fig. 4 and Fig. 5) in the revised manuscript to make the expression clearer. Moreover, we added Supplementary Note 1 and Supplementary 8 and 9 to describe the method we used for the background removal.

Similarly essential to their ability to image an entire fly brain is the use of ScaleView-A2 to clear the tissue, which was compatible with HMSiR, a blinking fluorophore that has been well described for single molecule localization. It is interesting and impactful to demonstrate suitable combinations of clearing solutions and blinking fluorophores. They should take care with their claims that super resolution has not been used within large, cleared tissues, and should acknowledge the contributions of Long et al. 2017 (doi:10.1038/nmeth.4309), who used Bessel beam SIM to localize single RNA molecules. Their use of BB-SIM on cleared tissue should be acknowledged, perhaps especially because in theory it should provide more optimal signal contrast.

Response: Thanks for the reviewer's reminder. For the work by Long et al. 2017 (doi:10.1038/nmeth.4309), we did cite the reference as reference 10 in the revised manuscript. In order to acknowledge their contributions, we have added more descriptions on page 3 in the revised manuscript as "Two recent studies have used a mounting medium with a high refractive-index (RI) to capture super-resolution images of neurons and mRNA molecules from the whole fly brain using Airyscan¹⁴ or Bessel Beam Structured Illumination Microscopy¹⁰, respectively. The optical resolution in these studies nonetheless was still limited to greater than 100 nm. Further, the dehydration caused by organic solvents in the mounting medium likely altered the structural integrity of the samples and the chemical properties of the labelled fluorophores¹⁵."

Note that there are still some limitations on BB-SIM (aberration-free system required; no double-resolution improved due to lower NA excitation and modulation) as mentioned in the previous answers.

While the authors have imaged an impressive and perhaps unprecedented level of detail across large spatial scales, there is little characterization of the biological structures that they have imaged and supposedly resolved. For such a large and heroic dataset and subsequent analysis, the reconstruction of 30 neurons ("The improvement in LLM-CT allows more than 30 neurons were traced and segmented") is underwhelming.

Response: Excellent lateral resolution (~ 30 nm) of LLM-CT allows us to perform manual segmentation of some but not all dopaminergic neurons in the whole fly brain. Though LLM-CT improves also axial resolution (~ 80 nm), dense neurites entangled at z direction remain indistinguishable. To further improve axial resolution, we are now working on combining LLM-CT with expansion microscopy and making tissue more transparent in a high refractive index clearing reagents.

Meanwhile, we added quantitative structural analysis of spatial distribution of Vesicular Monoamine Transporter (VMAT) proteins within a single neuron regulating learning and memory in the fly brain (Fig. 6, Supplementary Fig. 11). "Expression level and spatial distribution of proteins are essential to the function of a cell. Scientists historically quantify such endogenous proteins with immunostaining and visualize the label intensity at the cell body with confocal or wide-field microscopy to represent the expression level (**Fig. 6a**). Like many proteins, VMAT needs to be transported and/or locally synthesized at neuronal terminals to be functional and might be regulated locally by modulatory signals³¹, resulting in variable expression within a given neuron³². We explored this idea by visualizing VMAT molecules in a specific MB local neuron—the dorsal paired medial (DPM) neuron, located in the central brain, and calculated the density of VMAT protein along the different sub-regions in the MB.

We labeled VMAT with HMSiR and used our LLM-CT system to reconstruct VMAT locations. VMAT molecules along the MB lobes were classified into two groups: DPM positive (DPM+) and DPM negative (DPM-), based on the 3D digital intersection between VMAT and DPM (**Fig. 6b, 6e; Supplementary Movie 5**). Importantly, this classification could be accomplished only by using our LLM-CT method (**Fig. 6b**) and not by traditional Airyscan super-resolution confocal microscopy (**Fig. 6c**). To determine the imaging number of frames needed to map VMAT proteins (**Supplementary Fig. 10; Supplementary Note 2**), we determined theoretical resolution and localization density as discussed in Fig. 3. The

reliability of DPM+ VMAT localization was validated by targeting *VMAT^{RNAi}* expression to DPM neurons, which reduced the total number of VMAT molecules in DPM+ neurites but not in DPM- regions of MB lobes (**Fig. 6d, 6f; Methods**).

DPM neurites completely engulf both the vertical and horizontal lobes (axonal outputs) of the MBs³³ but activity in the vertical lobes appears to sub serve behavioral responses after a few sessions of olfactory classical conditioning³⁴. Furthermore, MB output neurons exhibit learning-gene dependent tuning plasticity³⁵ and synthesize new proteins during long-term memory formation^{36,37}. We hypothesized that VMAT in a DPM neuron might be distributed differentially to modulate functional differences among MB sectors. Accordingly, we manually segmented DPM neurites into 15 MB sectors: α 1, α 2, α 3, α' 1, α' 2, α' 3, β 1, β 2, β' 1, β' 2 and γ 1- γ 5, based on the pattern of input innervations from dopaminergic neurons (**Supplementary Fig. 11; Methods**). With quantitative single-molecule localization by LLM-CT, we found that VMAT proteins distribute unevenly within a DPM neuron and their density in each MB sector is highly variable among different flies (**Fig. 6g, 6h**). The plasticity of VMAT distribution we observe here may regulate the translation of transient MB activity into long-term memory coding at the MB outputs neurons.”

With editor and reviewer’s permission, we would like to add these results and related methods to the revised manuscript as an example of quantitative structural analysis.

Further, it was unclear to me how neurite reconstruction was performed, post processing with ThunderSTORM, so some citation or description of appropriate algorithms might be missing.

Response: We appreciate these constructive suggestions and have revised our method section accordingly. Briefly, the data processing is based on parallel running ThunderSTORM on a three-node Torque clusters. The pipeline of the post processing is depicted in Supplementary Fig. 7. For 3D visualization, the images are volume rendered in Avizo 9.4. The segmentation of single neurons is done manually using the Lasso tool in segmentation mode of Avizo 9.4.

I believe this manuscript has a value and novelty appropriate for Nature Communications, and with increased justification and clarity of the illumination parameters and some greater quantification of structural resolution within their biological samples, this manuscript could be accepted.

Erin Diel, PhD
Imaging Specialist, Harvard Center for Biological Imaging

We appreciate very much all critiques and constructive comments from the three reviewers. Thanks!

Reviewers' comments:

Reviewer #1 (Remarks to the Author):

The authors have made a major revision of the paper and have improved and corrected items of concern. I have no further comments.

Reviewer #2 (Remarks to the Author):

The reviewer believes that the authors could address all the mayor concerns and that the manuscript is suitable for a publication in Nature Communications after the minor concerns are addressed.

Minor concerns:

1) (Page 11 Line 261-265) "Recently, a combination of expansion microscopy and lattice lightsheet microscopy (ExLLSM) has achieved also nanometric resolution in the adult fly brain³⁸. For direct comparison between LLM-CT and ExLLSM, we imaged VMAT expression in the DPM neuron with in 4x expanded fly brain with lattice light sheet microscopy and compare the result with our data in Fig. 6 (Supplementary Fig. 12)." The statement should be moved to the results section, since it is a presented result.

2) (Page 12 Line 273-275) The statement that "tissue expansion with enzyme digestion in ExLLSM inevitably causes some protein loss during the lengthy sample preparation" should be rephrased since the removal of all proteins by tissue digestion and the imaging of the remaining fluorescent signal which is embedded in the remaining hydrogel is the trick of expansion microscopy.

3) An image of a cleared expanded *D. melanogaster* brain in comparison to a ScaleView-2A would be helpful to comprehend the both methods better.

4) Supplementary Figure 12 is very helpful. However, I would image the same entire structures with ExLLSM as with LLM-CT and compare those. Not just a fraction with the argument of tissue expansion (the working distance of the Olympus, XLPLN25XSVM2 should be sufficient).

Reviewer #3 (Remarks to the Author):

The authors have added acknowledgement to related work, some of which was published quite recently. It is impressive that they have also added comparisons across these related techniques, for example, the comparison of LLM-CT and ExLLSM. Further, their comparison between LLM-CT and Airyscan for colocalization analysis is an appreciated level of characterization of the biological applications that are possible with their technique. I now read their discussion as a thorough explanation of their contributions and important future considerations.

The authors have added an appreciated level of detail on their imaging setup, from listing part numbers to citing another publication by their group with additional information. Their added information in Supplemental Figure 1 is also a helpful description of lens-axicon combination and Bessel Beam lengths. There is a final clarifying point I have on which figures use which beam length:

Quoted from response from authors: Data presented in original Fig. 2a, 2b, 2c) are indeed obtained without astigmatism/cylindrical lens. Because of smaller field of view (~ 50 μm) (around $\frac{1}{4}$ camera chip size), we used the shorter lightsheet with less background contributed from the

sidelobe of Bessel beam. However, when a large field of view ($\sim 200 \mu\text{m}^2$) is needed for imaging the whole fly brain (shown in original Fig. 2d, 2e), a longer Bessel lightsheet is used and significant amount of sidelobe background is inevitable. To compensate the higher background induced by the sidelobe, a weak astigmatism was applied in the detection path to allow the exclusion of the out of focus signals in post-processing. In the revised manuscript, we separated original Fig. 2 into two figures (Fig. 4 and Fig. 5) to make the point clear.

Based on this description, I understand that figures 4 and 5 are using different techniques (with/without astigmatism). Therefore the imaging setup used should be added to figure legend or main text for clarification. The added text in discussion ("To minimize the deterioration of signal contrast, we used a weak astigmatism and excluded signals with asymmetric PSF during post-acquisition data processing (Figs. 4-6; Supplementary Note 1; Figs. 8, 9)") instead suggests that figures 4 and 5 both used long light sheet with astigmatism. Please clarify.

The revised manuscript is an improved version of what was previously seen and represents an impressive level of work and novel contribution to the field.

E. Diel

Response to Referees Letter

We appreciate all the positive recognitions and valuable suggestions from the reviewers. Following the suggestions, we have made necessary revisions, as well as additional comparison between our system and expansion microscopy. Attached please find our point-to-point responses, as well as the revised manuscript. To assist reading, the original comments are marked in black, our responses are in blue, and the revised sentences in the manuscript are in green.

Reviewers' comments:

Reviewer #1 (Remarks to the Author):

The authors have made a major revision of the paper and have improved and corrected items of concern. I have no further comments.

We thank the suggestions from the reviewer, which completes the manuscript.

Reviewer #2 (Remarks to the Author):

The reviewer believes that the authors could address all the mayor concerns and that the manuscript is suitable for a publication in Nature Communications after the minor concerns are addressed.

Minor concerns:

1) (Page 11 Line 261-265) “Recently, a combination of expansion microscopy and lattice lightsheet microscopy (ExLLSM) has achieved also nanometric resolution in the adult fly brain³⁸. For direct comparison between LLM-CT and ExLLSM, we imaged VMAT expression in the DPM neuron with in 4x expanded fly brain with lattice light sheet microscopy and compare the result with our data in Fig. 6 (Supplementary Fig. 12).” The statement should be moved to the results section, since it is a presented result.

We have moved this section into the last part of the revised results section.

2) (Page 12 Line 273-275) The statement that “tissue expansion with enzyme digestion in ExLLSM inevitably causes some protein loss during the lengthy sample preparation” should be rephrased since the removal of all proteins by tissue digestion and the imaging of the remaining fluorescent signal which is embedded in the remaining hydrogel is the trick of expansion microscopy.

Thanks to reviewer’s suggestion. We have rephrased the sentence as follow in page 11:

For quantitative analysis of protein distribution, LLM-CT using standard immunostaining protocol is preferable (**Fig. 6f, 6h**). In ExLLSM, at equivalent high resolution (see

Supplementary Table 3), the quality of fluorescent signal in expanded hydrogel is too capricious for quantitative analysis of protein molecules (**Supplementary Fig. 12h, 12i**).

3) An image of a cleared expanded *D. melanogaster* brain in comparison to a ScaleView-2A would be helpful to comprehend the both methods better.

4) Supplementary Figure 12 is very helpful. However, I would image the same entire structures with ExLLSM as with LLM-CT and compare those. Not just a fraction with the argument of tissue expansion (the working distance of the Olympus, XLPLN25XSVMP2 should be sufficient).

We thank reviewer's suggestion. However, we are afraid that our lab is not technically capable yet to fully address these questions.

Gao et al mentioned in their ExLLSM paper (doi: 10.1126/science.aau8302) that imaging the entire expanded fly brain requires to image ~25,000 tiles using a fully automated lattice lightsheet system. They also developed a customized algorithm to run a specialized computer cluster in order to reduce the discontinuity occurring at the boundaries between image titles for stitching all datasets into an ensemble.

For a direct comparison between LLM-CT and ExLLSM, we replicated the ExLLSM microscopy system identical to Gao et al's optical setup. However, our system does not equip with the automated control and algorithm for imaging and processing ~25,000 tiles, respectively. Thus, using our ExLLSM, we addressed reviewer's suggestion by manually imaged and stitched 8 tiles of volumes covering all mushroom body lobes. We then compared image qualities taken by ExLLSM and LLM-CT, respectively (see below for the revised **Supplement Figure 12**).

Our results show that each super-resolution method has its own advantage. However, for quantitative analysis of protein distribution, we believe that LLM-CT using standard immunostaining protocol is preferable.

Supplementary Fig. 12. Comparison between images taken by LLM-CT and ExLLSM.

(a) A DPM neuron occupied around $150 \times 140 \times 100 \mu\text{m}^3$ volume in the fly brain (left). Expansion microscopy enlarged the entire DPM neuron 4X in x, y and z directions, 64 times in volume (right). (b, c) LLM-CT image of all VMAT molecules in the entire DPM neuron taken by a single optical volume. (d, e) ExLLSM image of the DPM neuron from stitching 8 tiles of volume image. (f, g) Enlarged volume images of the insets in c and e, respectively. Single-molecule localization of LLM-CT image (f) revealed numerous VMAT signals (orange) in DPM neurites (green) as well as VMAT signals outside DPM neurites (blue). Image resolution is comparable between ExLLSM (g) and LLM-CT (f), allowing reliable allocation of VMAT signals within or outside DPM neurites. (h, i) Enlarged images show significantly more DPM+ VMAT molecules captured with LLM-CT than ExLLSM. VMAT was immunostained with HMSiR (LLM-CT) or Alexa 635 (ExLLSM). Scale bars were indicated in the figure.

Reviewer #3 (Remarks to the Author):

The authors have added acknowledgement to related work, some of which was published quite recently. It is impressive that they have also added comparisons across these related techniques, for example, the comparison of LLM-CT and ExLLSM. Further, their comparison between LLM-CT and Airyscan for colocalization analysis is an appreciated level of characterization of the biological applications that are possible with their technique. I now read their discussion as a thorough explanation of their contributions and important future considerations.

The authors have added an appreciated level of detail on their imaging setup, from listing part numbers to citing another publication by their group with additional information. Their added information in Supplemental Figure 1 is also a helpful description of lens-axicon combination and Bessel Beam lengths. There is a final clarifying point I have on which figures use which beam length:

Quoted from response from authors: Data presented in original Fig. 2a, 2b, 2c) are indeed obtained without astigmatism/cylindrical lens. Because of smaller field of view ($\sim 50 \mu\text{m}$) (around $\frac{1}{4}$ camera chip size), we used the shorter lightsheet with less background contributed from the sidelobe of Bessel beam. However, when a large field of view ($\sim 200 \mu\text{m}^2$) is needed for imaging the whole fly brain (shown in original Fig. 2d, 2e), a longer Bessel lightsheet is used and significant amount of sidelobe background is inevitable. To compensate the higher background induced by the sidelobe, a weak astigmatism was applied in the detection path to allow the exclusion of the out of focus signals in post-processing. In the revised manuscript, we separated original Fig. 2 into two figures (Fig. 4 and Fig. 5) to make the point clear.

Based on this description, I understand that figures 4 and 5 are using different techniques (with/without astigmatism). Therefore the imaging setup used should be added to figure legend or main text for clarification. The added text in discussion (“To minimize the deterioration of signal contrast, we used a weak astigmatism and excluded signals with asymmetric PSF during post-acquisition data processing (Figs. 4-6; Supplementary Note 1; Figs. 8, 9)”) instead suggests that figures 4 and 5 both used long light sheet with astigmatism. Please clarify.

The revised manuscript is an improved version of what was previously seen and represents an impressive level of work and novel contribution to the field.

E. Diel

Thanks very much for pointing out the inconsistency of the manuscript. The astigmatism-based post processing for background reduction is only applied to the data shown in Fig. 5 and 6. The reason is that the Bessel beam used here is longer than the beam used in the image acquisition for projection neurons (Fig. 2b,c,d, Fig. 3, Supplement Fig. 5b and Fig. 6), giant fiber neurons (Fig. 4b and Supplement Fig. 4) and for descending neurons (Fig. 4a). Since the generation of the Bessel beam with longer axial extent requires thinner annulus illumination pattern, the side lobes induced by the interference presented with the longer Bessel beam is apparently more significant.

We have revised discussion as followed (marked in green in the revised manuscript, page 12):

To minimize the deterioration of signal contrast, we used a weak astigmatism and excluded signals with asymmetric PSF during post-acquisition data processing (Figs. 5-6; Supplementary Note 1; Figs. 8, 9)

We have also added a new row in the table 1 to clarify if the astigmatism was used in each figure.

Image property	Fig. 2b,	Fig. 2c	Fig. 2d	Supplementary Fig. 4	Supplementary Fig. 5a	Fig. 3, Supplementary Fig. 5b, 6 (AL)	Supplementary Fig. 6 (LH)	Supplementary Fig. 6 (Calyx)	Fig. 4a	Fig. 4b	Fig. 5, Supplementary Fig. 9	Fig. 6b, d-h
Sample	MZ19-Gal4>UAS-mCD8::GFP	MZ19-Gal4>UAS-mCD8::GFP	MZ19-Gal4>UAS-mCD8::GFP	12862-Gal4>UAS-mCD8::GFP	MZ19-Gal4>UAS-mCD8::GFP	MZ19-Gal4>UAS-mCD8::GFP	MZ19-Gal4>UAS-mCD8::GFP	MZ19-Gal4>UAS-mCD8::GFP	Fru-Gal4>UAS-mCD8::GFP	12862-Gal4>UAS-Dscam::GFP	TH-Gal4>UAS-GCmMP6m	VT64246-Gal4>UAS-mCD8::GFP
HMSiR target	Anti-GFP, projection neurons	Anti-GFP, projection neurons	Anti-GFP, projection neurons	Anti-GFP, Giant fiber neurons	Anti-GFP, projection neurons	Anti-GFP, projection neurons	Anti-GFP, projection neurons	Anti-GFP, projection neurons	Anti-GFP, descending neurons	Anti-GFP, Dscam on giant fiber neurons	Anti-GFP, Dopamine neuron	Anti-VMAT, endogenous VMAT protein
Medium	PBS	ScaleView w-A2	ScaleView w-A2	PBS	PBS	ScaleView w-A2	ScaleView w-A2	ScaleView w-A2	ScaleView w-A2	ScaleView w-A2	ScaleView w-A2	ScaleView w-A2
Excitation wavelength (nm)	637	637	637	637	637	637	637	637	637	637	637	637
Lightsheet length (μm)	55	55	55	55	55	55	55	55	55	55	337	337
Excitation N.A. (outer; inner) of the excitation objective	0.26; 0.19	0.26; 0.19	0.26; 0.19	0.26; 0.19	0.26; 0.19	0.26; 0.19	0.26; 0.19	0.26; 0.19	0.26; 0.19	0.26; 0.19	0.19; 0.17	0.19; 0.17
Astigmatism	No	No	No	No	No	No	No	No	No	No	Yes	Yes
Voxel volume (x, y, z nm)	31 x 31 x 300	31 x 31 x 300	31 x 31 x 300	31 x 31 x 300	31 x 31 x 300	31 x 31 x 300	31 x 31 x 300	31 x 31 x 300	31 x 31 x 300	31 x 31 x 300	42 x 42 x 400	31 x 31 x 400
Image volume (x, y, z μm)	52 x 52 x 0.3	52 x 52 x 60	52 x 52 x 60	35 x 35 x 30	52 x 52 x 60	52 x 52 x 60	52 x 52 x 54	52 x 52 x 54	52 x 52 x 19	52 x 52 x 40	700 x 250 x 208	160 x 140 x 251
Number of slides	1	1	180	50,500	241,200	241,200	126,700	90,500	76,500	90,900	1,042,000	175,700
Number of subunit	1 x 1 x 1	1 x 1 x 1	1 x 1 x 1	1 x 1 x 1	1 x 1 x 1	1 x 1 x 1	1 x 1 x 1	1 x 1 x 1	1 x 1 x 1	1 x 1 x 1	4 x 1 x 1	1 x 1 x 1
Subunit size (x, y, z μm)	52 x 52 x 0.3	52 x 52 x 0.3	52 x 52 x 60	35 x 35 x 30	52 x 52 x 60	52 x 52 x 60	52 x 52 x 54	52 x 52 x 54	52 x 52 x 19	52 x 52 x 40	175 x 250 x 208	160 x 140 x 251
Pixel number per frame (x, y)	512 x 512	512 x 512	512 x 512	352 x 352	512 x 512	512 x 512	512 x 512	512 x 512	512 x 512	512 x 512	1400 x 2000	1600 x 1400
Exposure time/frame (ms)	100	100	100	100	100	100	100	100	100	100	80	80
Volume speed ($\mu\text{m}^3/\text{sec}$)	N.A.	N.A.	N.A.	3,675	8,112	8,112	8,112	8,112	8,112	8,112	700,000	89,600
Imaging period (hr)	100 ms	100 ms	100 ms	1.4	6.7	6.7	3.5	2.5	1.7	2.5	23.2	4.8
Localized molecules	N.A.	N.A.	100,914	1,415,065	N.A.	26,714,852	6,407,961	4,015,525	36,729,117	722,594	523,273,389	(Average)
Corresponding movie	N.A.	N.A.	N.A.	N.A.	N.A.	Supplementary Movie 1, 2	Supplementary Movie 2, 3	Supplementary Movie 2	N.A.	N.A.	Supplementary Movie 4	Supplementary Movie 5

1. Gao R, *et al.* Cortical column and whole-brain imaging with molecular contrast and nanoscale resolution. *Science* **363**, (2019).

REVIEWERS' COMMENTS:

Reviewer #2 (Remarks to the Author):

The authors have addressed all raised points in the paper and have improved and corrected items of concern. I have no further comments.